# Towards Lossless Memory-efficient Training of Spiking Neural Networks via Gradient Checkpointing and Spike Compression

**Yifan Huang**[1,2,3]**, Wei Fang**[4,*]**, Zecheng Hao**[1,2,5]**, Zhengyu Ma**[3]**, Yonghong Tian**[1,2,3,4,*]

[1]School of Computer Science, Peking University, Beijing, China
[2]Beijing Key Laboratory of Brain-inspired Spiking Large Models, Beijing, China
[3]Peng Cheng Laboratory, Shenzhen, China
[4]School of Electronic and Computer Engineering, Shenzhen Graduate School,
Peking University, Shenzhen, China
[5]State Key Laboratory for Multimedia Information Processing, Beijing, China
`{yfhuang,fwei,haozecheng,yhtian}@pku.edu.cn`, `mazhy@pcl.ac.cn`

## Abstract

Deep spiking neural networks (SNNs) hold immense promise for low-power event-driven computing, but their direct training via backpropagation through time (BPTT) incurs prohibitive memory cost, which limits their scalability. Existing memory-saving approaches, such as online learning, BPTT-to-BP, and reversible networks, compromise accuracy, training speed, or applicability. In this work, we propose a novel and broadly applicable pipeline for memory-efficient SNN training that preserves BPTT's accuracy. Our pipeline integrates layer-wise gradient checkpointing with lossless spike compression to eliminate internal state storage and reduce the memory cost of per-layer input spikes. We also introduce a multi-stage checkpoint adjustment strategy that adaptively refines checkpoint placement based on profiling results to further optimize memory usage and improve training speed. Wrapped in an optimization pass, the pipeline automatically restructures the computation flow before training with minimal user effort. Extensive experiments on diverse architectures and tasks demonstrate up to $8\times$ memory efficiency gains with $\leq 20\%$ speed reduction and no accuracy loss. Our method provides a practical solution for efficient and scalable SNN training. Code is available at `https://github.com/AllenYolk/snn-gradient-checkpointing`.

## 1 Introduction

Inspired by the dynamics of biological neurons (Gerstner et al., 2014), spiking neural networks (SNNs) have emerged as the third generation of neural network models (Maass, 1997). SNNs transmit information via discrete spikes rather than continuous activations in conventional artificial neural networks (ANNs). Their sparse and event-driven nature makes them ideal for deployment on neuromorphic chips (Merolla et al., 2014; Akopyan et al., 2015; Davies et al., 2018; Pei et al., 2019) for inference, offering significant potential for low-power edge computing (Yao et al., 2024). To train a deep SNN end-to-end, the temporal dimension is discretized into $T$ time steps so that the SNN can be considered as a binary-activated recurrent neural network (RNN) (Fang et al., 2023a; Eshraghian et al., 2023). Then, backpropagation through time (BPTT) (Werbos, 1990) is adopted to compute parameter updates, with surrogate gradient (SG) tackling the non-differentiable spike emission process (Neftci et al., 2019; Wu et al., 2018; Shrestha & Orchard, 2018). With the BPTT-based framework, low-latency deep SNNs can be directly trained using powerful graphics processing units (GPUs) (Chetlur et al., 2014) and yield competitive performance (Yao et al., 2025; Wang et al., 2024; Lv et al., 2024a; Chen et al., 2025).

Despite its high accuracy and broad applicability, BPTT imposes intensive memory overhead (Meng et al., 2023). For an $L$-layer SNN unfolded over $T$ time steps, BPTT requires $\mathcal{O}(LT)$ memory to store intermediate states, compared to $\mathcal{O}(L)$ for a structurally similar ANN. Consequently, SNN direct training is more likely to exceed the memory capacity of computational devices. The scaling of SNNs to deeper architectures and more time steps is thus severely hindered.

---

*Corresponding authors.

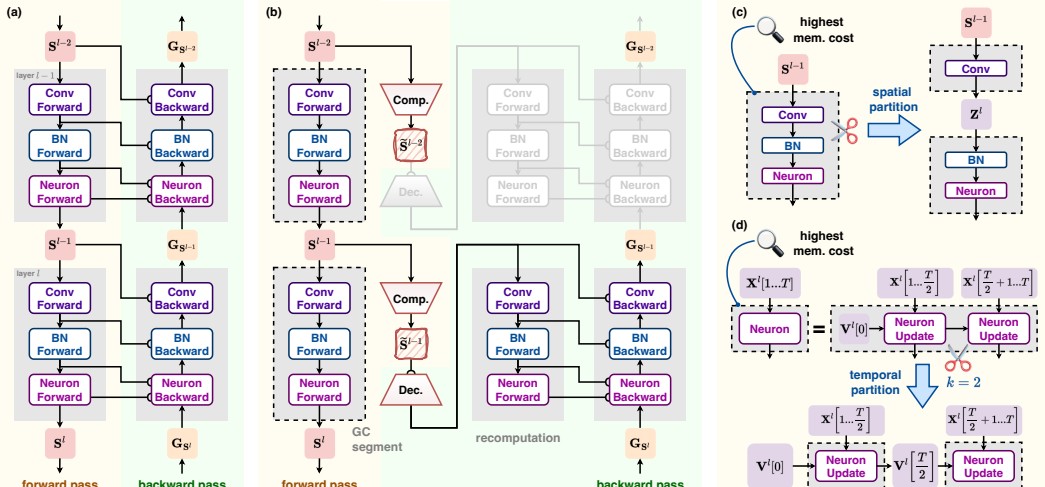

Figure 1: Comparison of (a) BPTT and (b) gradient checkpointing with spike compression. We use grey boxes with dashed borders to denote gradient checkpointing segments. (c) Spatial segment partitioning. (d) Temporal segment partitioning.

Several approaches have been explored to reduce the memory demands of BPTT-based SNN training, including online learning (Bellec et al., 2020; Xiao et al., 2022; Meng et al., 2023; Yin et al., 2023; Jiang et al., 2024), BPTT-to-BP (Xiao et al., 2021; Wu et al., 2023; Kheradpisheh et al., 2022; Yu et al., 2024), and reversible networks (Zhang & Zhang, 2024; Hu et al., 2024). However, these methods compromise training speed, accuracy, or generality across SNN models (see Section 2.2 and Table 5 for details). Also, their implementations require manual architectural modifications or training code rewrites, which are error-prone and cumbersome. These limitations highlight the need for a broadly applicable and user-friendly solution that improves the memory efficiency of SNN direct training while preserving training speed and performance.

In this work, we propose an automatic pipeline that combines gradient checkpointing (Chen et al., 2016) and spike compression to address the challenge (Figure 1). Our analysis identifies internal states and per-layer input spikes as the dominant memory consumers in SNN training. To this end, we employ layer-wise gradient checkpointing to eliminate internal state storage, and losslessly compress the input spikes before saving them to reduce their memory footprint. To further optimize peak memory usage, we insert additional checkpoints spatio-temporally into high-cost layers. Check-pointed segments with no benefit on peak memory are then greedily reverted to standard BPTT segments to accelerate training. The entire process is encapsulated in an optimization pass that automatically reconfigures the computation flow before training, requiring minimal user intervention. The proposed method obtains up to $8\times$ memory efficiency gains with an affordable training speed drop and preserved accuracy on extensive experiments. Our main contributions are:

**(1) Memory cost analysis.** We analyze the memory cost of SNN direct training and identify input spikes and internal states as primary memory consumers.

**(2) An automatic pipeline.** We propose a broadly applicable pipeline that integrates gradient checkpointing with spike compression for memory-efficient SNN training.

**(3) Efficiency and Accuracy.** We obtain substantial memory savings on diverse SNN models and task settings with acceptable speed trade-offs and maintained accuracy.

## 2 RELATED WORK

### 2.1 BPTT-BASED SNN DIRECT TRAINING

If simulated on discrete time steps, SNNs can be trained end-to-end as binary-activated RNNs through BPTT (Werbos, 1990), with SG addressing the non-differentiability of the spike firing process (Neftci et al., 2019; Wu et al., 2018; Shrestha & Orchard, 2018). Compared to ANN-to-SNN conversion

(Cao et al., 2015; Bu et al., 2022; Hu et al., 2023; Hao et al., 2023b;a), this approach enables low inference latency (Wu et al., 2019) and broader task applicability, thus attracting increasing attention. Recent advancements have improved the performance of SNN direct training by adapting ANN architectures like ResNet (He et al., 2016) and Transformer (Vaswani et al., 2017; Dosovitskiy et al., 2021) to spiking ResNets (Fang et al., 2021a; Hu et al., 2025) and spiking Transformers (Zhou et al., 2023; Yao et al., 2023; Zhou et al., 2024). Other works enhance neuron models (Fang et al., 2021b; Yao et al., 2022; Fang et al., 2023b; Huang et al., 2024a; Li et al., 2024b; Huang et al., 2024b). For instance, the parallel spiking neuron (PSN) family (Fang et al., 2023b) models neuronal dynamics as a linear projection of the input over time, enabling temporal parallelization and efficient capturing of long-term dependencies. Despite the advances in performance, the memory overhead of BPTT remains a key bottleneck.

## 2.2 Memory-efficient SNN Direct Training

BPTT's $\mathcal{O}(LT)$ memory complexity motivates methods to reduce SNN training memory usage. **Online learning** (Bellec et al., 2020; Xiao et al., 2022; Meng et al., 2023; Yin et al., 2023; Bohnstingl et al., 2023; Jiang et al., 2024) truncates temporal gradients and stores only the intermediate results at the current step. However, the gradient mismatch results in a severe performance drop on temporal tasks. Its step-wise running mode undermines its compatibility with widely adopted temporal parallelization techniques like PSN (Fang et al., 2023b). **BPTT-to-BP approximation** (Xiao et al., 2021; Wu et al., 2023; Kheradpisheh et al., 2022; Yu et al., 2024) trains an SNN by backpropagating through a static proxy based on firing rates, effectively removing the temporal gradient dimension. Despite its memory and time efficiency, BPTT-to-BP can hardly handle sequential data due to the neglect of temporal information, thus limiting its applicability. Last but not least, **reversible networks** (Gomez et al., 2017; Zhang & Zhang, 2024; Hu et al., 2024) reconstruct intermediate features reversely during backward pass rather than storing them. It preserves BPTT-level accuracy, but imposes strict architectural constraints and significantly slows training. In conclusion, existing methods trade off accuracy, speed, or applicability; they also require manual modifications on model architectures and training codes. In contrast, our pipeline reduces memory usage with an affordable extra time cost, maintains accuracy and broad compatibility, and demands minimal user effort.

## 3 Preliminaries

### 3.1 Spiking Neural Networks

SNNs can be regarded as ANNs augmented with bio-inspired spiking neuronal dynamics (Li et al., 2024a). To train an SNN directly, its dynamics are simulated on $T$ discrete time steps, and the spike signals are represented as binary activations. For example, the discrete-time dynamics of a $L$-layer SNN composed of leaky integrate-and-fire (LIF) neurons (Gerstner et al., 2014) can be described as

$$
\begin{aligned}
\mathbf{X}^l[t] &= g^l(\mathbf{S}^{l-1}[t]; \mathbf{W}^l), \\
\mathbf{H}^l[t] &= \lambda \mathbf{V}^l[t-1] + \mathbf{X}^l[t], \\
\mathbf{S}^l[t] &= \Theta(\mathbf{H}^l[t] - V_{\text{th}}), \\
\mathbf{V}^l[t] &= \mathbf{H}^l[t](1 - \mathbf{S}^l[t]).
\end{aligned}
\tag{1}
$$

Equation (1). Here, $l \in \{1, \ldots, L\}$ is the layer index, and $t \in \{1, \ldots, T\}$ is the time step index. $\mathbf{X}$ is the input current, $\mathbf{H}$ and $\mathbf{V}$ are the membrane potentials before and after spike emission, and $\mathbf{S}$ is the output spike (a.k.a. activation). $\mathbf{X}^l[t]$ can be computed from the previous layer's output $\mathbf{S}^{l-1}[t]$ via a linear transformation $g^l$ with weight $\mathbf{W}^l$ (bias is omitted). $\lambda \in (0, 1)$ is the decay factor, $V_{\text{th}} > 0$ is the firing threshold, and $\Theta(x)$ is the Heaviside step function (yields 1 if $x \geq 0$ and 0 otherwise). The elements of $\mathbf{S}^l$ $(l > 0)$ are either 0 (no spike) or 1 (spike), while the network input $\mathbf{S}^0$ is not necessarily binary (Rathi & Roy, 2023). Notice that the second to fourth lines of Equation (1) are element-wise, and secondary neuronal parameters like the reset and resting potentials are omitted for simplicity. We use LIF as the default neuron model throughout this work.

### 3.2 Gradient Checkpointing

Gradient checkpointing (GC) (Chen et al., 2016) was originally proposed for ANN training to trade computation for memory. Standard backpropagation stores all intermediate results for gradient computation, as Figure 1(a) shows. By contrast, GC stores merely a subset of activations (a.k.a. **checkpoints**) and discards the others; the network is thus divided into several **GC segments**, each

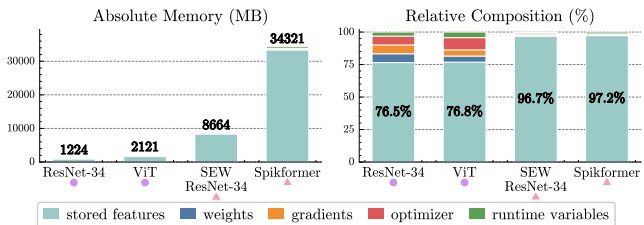

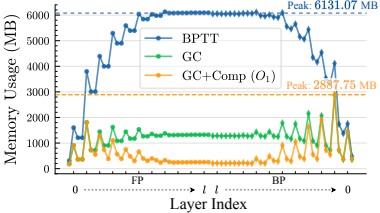

Figure 2: The memory cost breakdown of ANNs and SNNs when peak memory consumption is reached during training on ImageNet (see Appendix A).

Figure 3: The memory cost evolution when training a Spiking VGG on CIFAR10-DVS. Dashed lines indicate peak memory consumptions.

saving only its input. During backward pass on a segment, the forward computation is rerun from the segment's checkpointed input to restore the dropped activations needed for calculating gradients, as illustrated in Figure 1(b). Since forward pass is far less costly than backward pass, GC's extra time cost is affordable. GC has been successfully applied to temporal models like recurrent neural networks (Gruslys et al., 2016) and neural ordinary differential equations (Zhuang et al., 2020) to reduce training memory cost.

Previous studies have explored applying GC along the temporal dimension of SNNs (Singh et al., 2022; Bencheikh et al., 2024), achieving notable memory savings on shallow networks with a large $T$ ($T \geq 100$). However, these approaches do not consider the spatial dimension, and have not been evaluated on advanced larger-scale SNNs that typically adopt short time horizons ($T \leq 16$). In addition, they lack an automated, user-friendly GC workflow, which limits ease of use in practice.

## 4 METHODS

### 4.1 MEMORY COST ANALYSIS OF BPTT-BASED SNN DIRECT TRAINING

In BPTT-based SNN direct training, memory usage primarily stems from: (1) model **parameters**, (2) **gradients**, (3) **optimizer states**, (4) intermediate features, including each layer's **input** and **internal states**, that are stored during forward pass for backward gradient computation, and (5) **temporary runtime variables** dynamically allocated and immediately freed. An upper bound for the peak memory can be formulated as:

$$\mathcal{M}_{\text{BPTT}}^{\text{peak}} \leq \sum_l \left( \mathcal{M}_{\mathbf{W}^l} + \mathcal{M}_{\mathbf{G}^l} + \mathcal{M}_{\Lambda^l} + \mathcal{M}_{\mathbf{S}^{l-1}} + \mathcal{M}_{\Omega^l} \right) + \max_l \mathcal{M}_{\mathbf{R}^l}, \tag{2}$$

where $\mathcal{M}_{\mathbf{W}^l}$, $\mathcal{M}_{\mathbf{G}^l}$, $\mathcal{M}_{\Lambda^l}$, $\mathcal{M}_{\mathbf{S}^{l-1}}$, $\mathcal{M}_{\Omega^l}$, and $\mathcal{M}_{\mathbf{R}^l}$ are the memory consumptions of the weights, gradients, optimizer states, inputs, internal states, and runtime variables at layer $l$, respectively.

A key feature of SNN direct training is that intermediate features (inputs and internal states) dominate memory usage. As shown in Figure 2, for ResNet-34 (He et al., 2016) and ViT (Dosovitskiy et al., 2021) trained on ImageNet (Deng et al., 2009), intermediate features occupy about 77% of the memory at peak usage. In contrast, for their SNN counterparts with $T = 4$, the ratios rise to over 96%. This is because SNNs' $T$ time steps scale intermediate feature sizes by $\mathcal{O}(T)$, while the sizes of weights, gradients and optimizer states stay unchanged. Therefore, memory optimization for SNN direct training should prioritize reducing internal states and input spike storage at each layer.

### 4.2 LAYER-WISE GRADIENT CHECKPOINTING

In standard BPTT, all internal states must be stored, resulting in a memory cost of up to $\sum_l \mathcal{M}_{\Omega^l}$. To reduce this cost, we apply GC (Chen et al., 2016) to each layer $l \in \{1, \dots, L\}$. During the forward pass on layer $l$, only the input $\mathbf{S}^{l-1}$ and weight $\mathbf{W}^l$ are stored. In the backward pass, internal states $\Omega^l$ are reconstructed through an extra local forward pass given $\mathbf{S}^{l-1}$ and $\mathbf{W}^l$. With $\mathbf{S}^{l-1}$, $\mathbf{W}^l$ and $\Omega^l$, we can propagate the gradients back through layer $l$, as Figure 1(b) shows.

With GC, $\Omega^l$ is allocated and freed during layer $l$'s backward pass. Thus, at most one layer's internal states are stored in memory at any given time. The peak memory's upper bound then becomes:

$$\mathcal{M}_{\mathrm{GC}}^{\mathrm{peak}} \leq \sum_l \left( \mathcal{M}_{\mathbf{W}^l} + \mathcal{M}_{\mathbf{G}^l} + \mathcal{M}_{\Lambda^l} + \mathcal{M}_{\mathbf{S}^{l-1}} + \cancel{\mathcal{M}_{\Omega^l}} \right) + \max_l \left( \cancel{\mathcal{M}_{\Omega^l}} + \mathcal{M}_{\mathbf{R}^l} \right). \quad (3)$$

Since internal states in SNNs consume far more memory than in ANNs (Figure 2), GC's effectiveness will be more pronounced in SNNs compared to ANNs.

### 4.3 LOSSLESS INPUT SPIKE COMPRESSION

Input spikes $\mathbf{S}^{l-1}$ must be stored even if GC is applied. For most SNN programming frameworks (Fang et al., 2023a; Eshraghian et al., 2023), spikes are represented as 32-bit floats (or 16-bit with automatic mixed precision) for compatibility with arithmetic operations. However, 32-bit storage is redundant for binary values. Therefore, instead of storing $\mathbf{S}^{l-1}$ as floats during forward pass, we store its compressed form $\tilde{\mathbf{S}}^{l-1}$, as Figure 1(b) shows. $\tilde{\mathbf{S}}^{l-1}$ is decompressed to $\mathbf{S}^{l-1}$ when needed in backward pass (Algorithm 1). The peak memory's upper bound then becomes:

$$\mathcal{M}_{\mathrm{GC+Comp}}^{\mathrm{peak}} \leq \sum_l \left( \mathcal{M}_{\mathbf{W}^l} + \mathcal{M}_{\mathbf{G}^l} + \mathcal{M}_{\Lambda^l} + \cancel{\mathcal{M}_{\mathbf{S}^{l-1}}} + \mathcal{M}_{\tilde{\mathbf{S}}^{l-1}} \right) + \max_l \left( \mathcal{M}_{\Omega^l} + \mathcal{M}_{\mathbf{R}^l} \right). \quad (4)$$

The spike compressor must be lossless to ensure computational equivalence with standard BPTT. For instance, **bit representation** uses 1 bit per binary value, achieving up to $32\times$ compression over 32-bit floats. Alternatives include **sparse representation** that records the indices of non-zero elements, and **lossless bit stream compressors** like Zstandard (Collet & Kucherawy, 2018) and asymmetric numeral systems (ANS) (Duda, 2013). While bit representation cannot benefit from spike sparsity, it is faster and more memory-saving than the alternatives in most cases (see Appendix M). Hence, we choose it by default. To further accelerate compression and decompression, we handcraft Triton kernels (Tillet et al., 2019). Notice that compression is skipped for non-binary inputs (e.g., $\mathbf{S}^0$).

### 4.4 ADJUSTING GRADIENT CHECKPOINTING STRUCTURE

Figure 3 depicts the memory evolution during a training iteration of a Spiking VGG on CIFAR10-DVS (see Appendix B for explanations). For standard BPTT (blue), the peak occurs in deep layers during backward pass. Layer-wise GC (green) reduces deep-layer memory, shifting the peak to shallower layers, and spike compression (orange) further lowers deep-layer cost. After these optimizations, the global peak memory $\mathcal{M}^{\mathrm{peak}}$ achieved at the critical layer far exceeds the local peaks elsewhere. Notice that model trainability on specific devices depends only on this global peak. This motivates us to adjust the GC structure to further enhance global efficiency by allowing slightly higher memory usage in non-critical layers. We propose three strategies accordingly and summarize them in Algorithm 2.

**Spatial Segment Partitioning** To reduce $\mathcal{M}^{\mathrm{peak}}$, we first identify the GC segment $l^*$ with the largest peak memory and then insert a spatial checkpoint within it. In other words, we split $l^*$ along the layer dimension into two **spatial subsegments** $l_1^*$ and $l_2^*$, as Figure 1(c) shows. The spatial partition point is defined by the user (see Appendix I). Since $\mathcal{M}_{\Omega^{l^*}} > \max\{\mathcal{M}_{\Omega^{l_1^*}}, \mathcal{M}_{\Omega^{l_2^*}}\}$, a reduction of $\max_l \mathcal{M}_{\Omega^l}$ is guaranteed. However, $\mathcal{M}^{\mathrm{peak}}$ may not drop due to the added checkpoint. This process repeats until $\mathcal{M}^{\mathrm{peak}}$ cannot further decrease.

**Temporal Segment Partitioning** Temporal partitioning similarly finds the critical segment $l^*$ and splits it along the time axis into $k$ sequential **temporal subsegments**, as shown in Figure 1(d). Each temporal subsegment checkpoints both its inputs and initial hidden states to enable recomputation during backward pass. Users should set the temporal partitioning factor $k$ and define the state transition function (see Appendix I). The procedure repeats until $\mathcal{M}^{\mathrm{peak}}$ cannot be further reduced. Temporal partitioning is applied conservatively after spatial partitioning as a complementary strategy, since splitting segments along time disables temporal parallelism and limits temporal kernel fusion, resulting in restricted applicability and slower training.

**Greedy Segment Restoration** For GC segments whose local memory cost is well below $\mathcal{M}^{\mathrm{peak}}$, we can safely revert them to standard BPTT blocks (i.e., storing all intermediate features) without

**Algorithm 1** One iteration of SNN training with layer-wise GC and spike compression.

---

**Input:** parameters $\{\mathbf{W}^l\}_{l=1}^L$; network input $\mathbf{S}^0$; compressor $C(\cdot)$; other hyperparameters.
**Output:** trained parameters $\{\mathbf{W}^l\}_{l=1}^L$.

1: *// forward pass*
2: **for** $l = 1, 2, \ldots, L$ **do**
3:     $\mathbf{S}^l \leftarrow \text{layer}^l(\mathbf{S}^{l-1}; \text{autograd} = \text{False})$;
4:     **if** $\mathbf{S}^{l-1}$ is binary **then**
5:         Compress: $\tilde{\mathbf{S}}^{l-1} \leftarrow C(\mathbf{S}^{l-1})$;
6:         Save $\tilde{\mathbf{S}}^{l-1}$, and free $\mathbf{S}^{l-1}$;
7:     **else**
8:         Save $\mathbf{S}^{l-1}$;
9:     **end if**
10: **end for**
11: Compute the loss $\mathcal{L}$ and the gradient $\frac{\partial \mathcal{L}}{\partial \mathbf{S}^L}$;
12: *// backward pass*
13: **for** $l = L, L-1, \ldots, 1$ **do**
14:     **if** $\mathbf{S}^{l-1}$ is compressed **then**
15:         Decompress: $\mathbf{S}^{l-1} \leftarrow C^{-1}(\tilde{\mathbf{S}}^{l-1})$;
16:     **end if**
17:     $\mathbf{S}^l \leftarrow \text{layer}^l(\mathbf{S}^{l-1}; \text{autograd} = \text{True})$;
18:     Compute $\frac{\partial \mathcal{L}}{\partial \mathbf{W}^l}, \frac{\partial \mathcal{L}}{\partial \mathbf{S}^{l-1}}$ by BPTT;
19:     Free the saved tensors of layer $l$;
20: **end for**
21: Update the parameters $\{\mathbf{W}^l\}_{l=1}^L$.

---

**Algorithm 2** GC structure adjustment.

---

**Input:** A list of GC segments $\Psi = [\text{seg}^l]_{l=1}^L$.
**Output:** the adjusted GC segment list.

1: *// spatial partitioning*
2: **while** True **do**
3:     Find $l^* = \arg\max_l(\mathcal{M}_l^{\text{peak}})$;
4:     Spatially split: $\text{seg}^{l^*} \rightarrow \{\text{seg}^{l^*_1}, \text{seg}^{l^*_2}\}$
5:     **if** global $\mathcal{M}^{\text{peak}}$ doesn't decrease **then**
6:         Revert the split; break;
7:     **end if**
8: **end while**
9: *// temporal partitioning*
10: **while** True **do**
11:     Find $l^* = \arg\max_l(\mathcal{M}_l^{\text{peak}})$;
12:     Temporally split: $\text{seg}^{l^*} \rightarrow \{\text{seg}^{l^*_i}\}_{i=1}^k$;
13:     **if** global $\mathcal{M}^{\text{peak}}$ doesn't decrease **then**
14:         Revert the split; break;
15:     **end if**
16: **end while**
17: *// greedy restoration*
18: sort $\Psi$ descendingly by forward time cost;
19: **for** $\text{seg}^l$ in $\Psi$ **do**
20:     Restore $\text{seg}^l$ to a BPTT segment;
21:     **if** global $\mathcal{M}^{\text{peak}}$ increases **then**
22:         Re-enable GC for $\text{seg}^l$;
23:     **end if**
24: **end for**

---

increasing $\mathcal{M}^{\text{peak}}$. Since GC segments require an extra forward pass for recomputation, restoring them accelerates training. Specifically, we first profile the forward time cost of each GC segment, and then greedily restore the segments with the largest time cost. The change is kept only if $\mathcal{M}^{\text{peak}}$ does not increase. This process terminates after all segments are considered.

## 4.5 AUTOMATIC PIPELINE

To minimize user intervention, we wrap all the above strategies into an automatic pipeline. Users can set the *level* parameter to specify the applied strategy set. At level $O_1$, only layer-wise GC and spike compression are enabled; $O_2$ additionally applies spatial segment partitioning; $O_3$ further incorporates temporal partitioning; $O_4$ additionally activates greedy segment restoration. Default settings cover most cases, while advanced users can customize spatio-temporal partition schemes. This design balances simplicity and extensibility.

```
net = memory_optimization(
    net,
    (Conv1dBNNeuron, Conv2dBNNeuron, QKACore, SSACore),
    dummy_input=torch.rand(32, 3, 224, 224),
    compress_x=True,
    level=4,
    verbose=True,
    temporal_split_factor=2,
)
```

Figure 4: The pipeline's user interface.

## 4.6 MEMORY-EFFICIENT LIF KERNEL

Beyond the optimization pipeline, kernel-level improvements can bring further efficiency gains. We therefore design a Triton kernel (Tillet et al., 2019) for the widely adopted LIF neuron. The BPTT formulation of LIF can be derived from Equation (1) as:

$$\frac{\partial \mathcal{L}}{\partial \mathbf{X}^l[t]} = \left( \frac{\partial \mathcal{L}}{\partial \mathbf{S}^l[t]} - \frac{\partial \mathcal{L}}{\partial \mathbf{V}^l[t]} \mathbf{H}^l[t] \right) \Theta'_{\text{sg}}(\mathbf{H}^l[t] - V_{\text{th}}) + \frac{\partial \mathcal{L}}{\partial \mathbf{V}^l[t]}(1 - \mathbf{S}^l[t]),$$
$$\frac{\partial \mathcal{L}}{\partial \mathbf{V}^l[t-1]} = \lambda \frac{\partial \mathcal{L}}{\partial \mathbf{X}^l[t]}, \tag{5}$$

Table 1: Comparison of training speed and memory cost. The throughput and memory cost ratios relative to "SJLIF, BPTT" are shown in parentheses.

| Task | T | Batch Size | Network | LIF impl. | Method | Throughput (sample / s) ↑ | Peak Alloc. Mem. (MB) ↓ |
|------|---|-----------|---------|-----------|--------|--------------------------|-------------------------|
| Sequential CIFAR-10 | 32 | 128 | SCNN | SJLIF | BPTT | 4872.23 | 1317.23 |
| | | | | PTLIF | BPTT | 1054.68 | 1264.97 |
| | | | | MELIF | $O_4$ | 5138.76 (1.05×) | 474.98 (0.36×) |
| DVS128 Gesture | 16 | 16 | 7B-Net | SJLIF | BPTT | 114.52 | 8984.02 |
| | | | | PTLIF | BPTT | 36.52 | 8067.41 |
| | | | | MELIF | $O_4$ | 120.04 (1.05×) | 4213.86 (0.47×) |
| CIFAR10-DVS | 10 | 32 | Spiking VGG | SJLIF | BPTT | 290.26 | 6131.07 |
| | | | | PTLIF | BPTT | 150.69 | 5889.44 |
| | | | | MELIF | $O_4$ | 270.79 (0.93×) | 2349.39 (0.38×) |
| ImageNet | 4 | 32 | SEW ResNet-34 | SJLIF | BPTT | 309.04 | 8821.28 |
| | | | | PTLIF | BPTT | 202.83 | 7140.09 |
| | | | | MELIF | $O_4$ | 281.39 (0.91×) | 2004.14 (0.23×) |
| | | | Spikformer (8-512) | SJLIF | BPTT | 116.70 | 34264.76 |
| | | | | PTLIF | BPTT | 71.03 | 28779.13 |
| | | | | MELIF | $O_4$ | 93.58 (0.80×) | 7640.68 (0.22×) |
| | | | QKFormer (10-512) | SJLIF | BPTT | 86.15 | 44571.33 |
| | | | | PTLIF | BPTT | 55.65 | 37375.90 |
| | | | | MELIF | $O_4$ | 76.51 (0.89×) | 5219.93 (0.12×) |

where $\mathcal{L}$ is the loss and $\Theta'_{sg}$ is the surrogate gradient function. Accordingly, BPTT on a LIF layer requires storing only $\{\mathbf{H}^l[t]\}_{t=1}^T$ and $\{\mathbf{S}^l[t]\}_{t=1}^T$ during forward pass. We further avoid storing $\{\mathbf{S}^l[t]\}_{t=1}^T$ by reconstructing it during the LIF layer's backward pass through $\mathbf{S}^l[t] = \Theta(\mathbf{H}^l[t] - V_{th})$. In this way, the floating-point spikes can be dropped once their compression at the subsequent layer is done. We name the kernel as memory-efficient LIF (**MELIF**) and use it by default.

## 5 EXPERIMENTS

In this section, we evaluate the proposed method's memory efficiency, as well as training speed, compatibility, and accuracy. We also conduct case studies to highlight the importance of our method.

### 5.1 MEMORY COST AND TRAINING SPEED

We assess the memory and time cost of our method on Sequential CIFAR-10 (Fang et al., 2021b), DVS128 Gesture (Amir et al., 2017), CIFAR10-DVS (Li et al., 2017), and ImageNet (Deng et al., 2009). For ImageNet, we try three architectures: SEW ResNet-34 (Fang et al., 2021a), Spikformer (Zhou et al., 2023), and QKFormer (Zhou et al., 2024). See Appendix C for more details. As Table 1 shows, our memory optimization pipeline at $O_4$ combined with the Triton-based LIF kernel (MELIF) reduces the peak memory consumption to $0.12\times \sim 0.47\times$ of SNNs trained with standard BPTT using SpikingJelly's CuPy-based

Table 2: Ablation study of LIF implementation and optimization levels on CIFAR10-DVS.

| LIF impl. | Opt. Level | Throughput (sample / s) ↑ | Peak Alloc. Mem. (MB) ↓ |
|-----------|-----------|--------------------------|-------------------------|
| SJLIF | – | 290.26 | 6131.07 |
| PTLIF | – | 150.69 | 5889.44 |
| MELIF | – | 331.30 | 4865.06 |
| | $O_1$ | 246.81 | 2887.75 |
| | $O_3$ | 247.83 | 2349.39 |
| | $O_4$ | 270.79 | 2349.39 |

LIF (SJLIF). This great reduction in memory footprint is achieved with no or only a slight training slowdown ($\leq 20\%$; see Appendix K for a more detailed runtime decomposition). Table 2 shows that the proposed Triton kernel is significantly more memory- and time-efficient than SJLIF and the LIF in pure PyTorch (PTLIF). Moreover, layer-wise GC ($O_1$) and spatio-temporal GC segment partitioning ($O_3$) further reduce memory, while greedy restoration ($O_4$) mitigates the recomputation overhead of GC. A fine-grained ablation study on three GC adjustment strategies is provided in Appendix L.

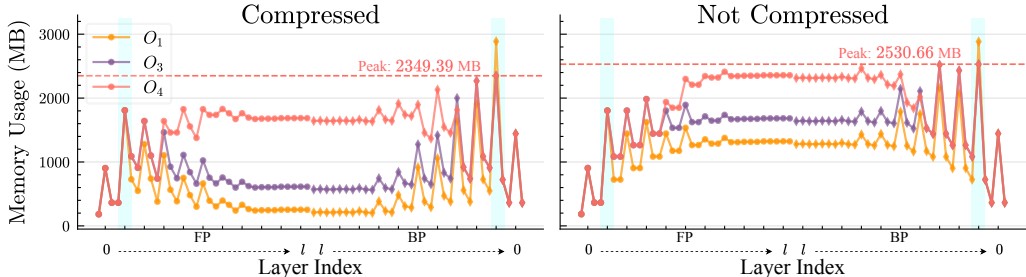

Figure 5: Spiking VGG memory evolution on CIFAR10-DVS under different optimization levels.

Table 3: Compatibility with temporally parallel SNNs.

| Task | Network | Neuron | Method | Peak Alloc. Mem. (MB) ↓ |
|------|---------|--------|--------|-------------------------|
| Sequential CIFAR-10 | SCNN | Sliding PSN | BPTT | 1302.69 |
| | | | $O_4$ | 599.34 (0.46×) |
| ImageNet | SEW ResNet-34 | PSN | BPTT | 7602.64 |
| | | | $O_4$ | 2544.28 (0.33×) |

Table 4: Compatibility with AMP and LOMO. Condition: ImageNet, QKFormer, MELIF, $O_4$.

| AMP? | LOMO? | Peak Alloc. Mem. (MB) ↓ |
|------|-------|-------------------------|
| ✗ | ✗ | 5219.93 |
| ✗ | ✓ | 5190.60 |
| ✓ | ✗ | 3158.02 |
| ✓ | ✓ | 3142.86 |

Finally, Figure 5 demonstrates that spike compression brings memory saving by providing more free space for GC structure adjustment (see Appendix J for a detailed discussion).

## 5.2 COMPATIBILITY WITH OTHER METHODS

Beyond LIF neurons, our method is compatible with other spiking neuron models. Table 3 shows that our approach effectively reduces memory usage for SNNs built with PSNs and Sliding PSNs (Fang et al., 2023b). Note that temporal parallelism is not compatible with BPTT-to-BP or online learning. Moreover, Table 4 demonstrates that our method can be seamlessly combined with prevalent memory-saving techniques, such as automatic mixed precision (AMP) (Micikevicius et al., 2018) and low-memory optimizer (LOMO) (Lv et al., 2024b) (see Appendix E for introductions).

## 5.3 MATHEMATICAL EQUIVALENCE WITH CONVENTIONAL BPTT

To verify that our pipeline produces unbiased gradients with respect to standard BPTT, we compare Sequential CIFAR-10 accuracies in Figure 6. The MELIF curves with and without $O_4$ optimization (green and orange) exactly overlap, showing that GC and spike compression do not introduce gradient bias. Their minor difference from the baseline (SJLIF, blue) stems from the different numerical behavior of Triton and CuPy. This gap is negligible, as the orange curve lies almost entirely within the baseline's error band. Additional results and discussion

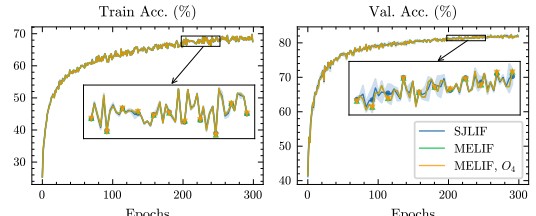

Figure 6: Sequential CIFAR-10 accuracies. SJLIF shows mean ± std over three runs, while the other two curves are single runs with a fixed seed.

on numerical discrepancies are provided in Appendices F and G. Overall, our pipeline preserves BPTT-level accuracy, which is its main advantage over other efficient training approaches.

## 5.4 COMPARISON WITH OTHER EFFICIENT TRAINING METHODS

Table 5 compares throughput, memory usage, gradient fidelity, and applicability constraints of representative efficient training methods. All methods use the same Spiking VGG model, except reversible networks, whose architectures are adjusted to match the VGG in parameter count (9.2 M)

Table 5: Comparison of SNN efficient training methods. Throughput and memory are tested on CIFAR10-DVS. 'Grad. Bias' indicates additional gradient approximation beyond surrogate gradients.

| Category | Method | Throughput (sample / s) ↑ | Peak Alloc. Mem. (MB) ↓ | Grad. Bias | Constraints |
|---|---|---|---|---|---|
| Vanilla | BPTT | 290.26 | 6131.07 | ✗ | ✗ |
| Online Learning | SLTT | 297.45 | 736.63 | ✓ | step-wise only |
| | OTTT | 216.78 | 969.21 | | |
| | NDOT | 168.48 | 1467.90 | | |
| BPTT-to-BP | Tandem SNN | 551.96 | 1706.68 | ✓ | no temporal dependency |
| | Rate-based | 497.07 | 1540.65 | | |
| Reversible Network | RevSResNet | 157.46 | 3198.78 | ✗ | reversible models only |
| | T-RevSNN | 191.36 | 1089.43 | | |
| Ours | $O_4$ | 270.79 | 2349.39 | ✗ | layer-wise only |

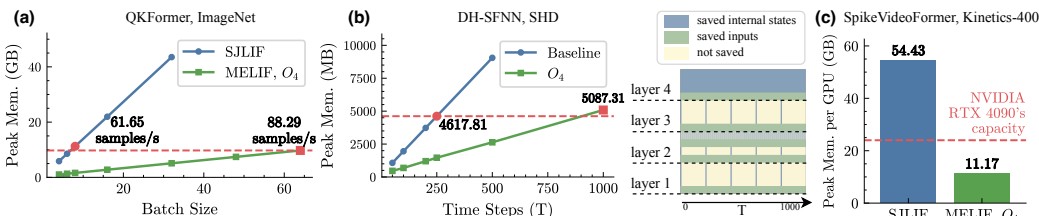

Figure 7: Case studies. The proposed pipeline enables (a) larger batch size, (b) finer temporal resolution, and (c) training large-scale SNNs on more accessible devices. The heatmap in (b) shows which intermediate features are saved during forward pass after $O_4$ optimization when $T = 1000$.

and feature-map resolution. Online learning methods like SLTT (Meng et al., 2023), OTTT (Xiao et al., 2022) and NDOT (Jiang et al., 2024) achieve the lowest memory cost but require step-wise execution, prohibiting techniques like temporal parallelism (Fang et al., 2023b) that are common in modern SNNs. BPTT-to-BP, such as Tandem SNN (Wu et al., 2023) and Rate-based BP (Yu et al., 2024), shows higher throughput but introduces substantial gradient bias, making it unsuitable for tasks with rich temporal dependencies. Reversible networks like RevSResNet (Zhang & Zhang, 2024) and T-RevSNN (Hu et al., 2024) reduce memory cost but significantly slow down training and impose strict architectural constraints. In contrast, our method balances speed and memory while maintaining mathematical equivalence to BPTT and supporting generic layer-wise SNNs.

## 5.5 CASE STUDIES

**QKFormer on ImageNet** Take QKFormer trained on ImageNet ($T = 4$) as an example. With our pipeline, the batch size can be increased by nearly $8\times$ without consuming more memory. Enlarging the batch size from 8 to 64 yields about $1.43\times$ training speedup, as shown in Figure 7(a).

**DH-SFNN on SHD** We evaluate our method on Spiking Heidelberg Digits (SHD) (Cramer et al., 2022) using DH-SFNN, a fully connected SNN ($700 \rightarrow 1024 \rightarrow 1024 \rightarrow 512 \rightarrow 20$) with dendritic heterogeneity LIF (DH-LIF) neurons (Zheng et al., 2024). Each DH-LIF contains four dendritic branches and a soma, resulting in five internal states per neuron. Batch size is set to 128. Existing efficient training approaches can hardly work here: online learning and BPTT-to-BP struggle with SHD's rich temporal dynamics, while reversible network is infeasible due to architectural constraints. In contrast, as Figure 7(b) shows, our method enables $4\times$ increase in $T$ with negligible extra memory cost, allowing finer temporal resolution and potentially better sequence modeling quality.

**SpikeVideoFormer on Kinetics-400** We train a SpikeVideoFormer (Zou et al., 2025) (55.9 M parameters) on Kinetics-400 (Kay et al., 2017) with $T = 32$ frames and $224 \times 224$ input resolution. Training with a batch size of 4 per GPU requires 54.43 GB of memory per device, restricting experiments to high-end hardware. Indeed, the original work uses eight A6000 GPUs, which is not affordable for many researchers. With our method, the peak memory per GPU is reduced to 11.17 GB,

enabling its training on widely accessible GPUs (e.g., 4090, 24 GB), as Figure 7(c) shows. This demonstrates that our approach can lower hardware barriers for cutting-edge SNN research.

## 6 CONCLUSION

In this work, we presented an automatic memory optimization pipeline for SNN direct training. The pipeline integrates layer-wise GC with lossless spike compression to reduce the memory footprint of intermediate features. We then adaptively adjust GC structure by spatio-temporal segment partitioning and greedy restoration to further reduce memory demand and GC's recomputation overhead. Experiments show that our pipeline achieves high memory efficiency while maintaining acceptable training speed, BPTT-level accuracy, and broad compatibility. This work provides a practical approach for efficiently training large-scale SNNs. Limitations and future directions are discussed in Appendix N.

## ACKNOWLEDGMENTS

This work was supported by the National Natural Science Foundation of China (62425101, 62332002), and Beijing Key Laboratory of Brain-inspired Spiking Large Models.

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

## A    DETAILS OF MEMORY COST BREAKDOWN

Figure 2 illustrates the memory breakdown of stored feature maps (input tensors and internal states of all layers), model weights, gradients, optimizer states, and temporary runtime variables when training SNNs or ANNs on ImageNet (Deng et al., 2009). We evaluate SEW ResNet-34 (Fang et al., 2021a) and Spikformer (Zhou et al., 2023) using the same settings as our main experiments (Appendix C), with ResNet-34 (He et al., 2016) and ViT (Dosovitskiy et al., 2021) mirroring the settings of SEW ResNet-34 and Spikformer, respectively. SEW ResNet-34 and Spikformer are implemented using SpikingJelly (Fang et al., 2023a), and the LIF model with CuPy backend is adopted; For ResNet-34 and ViT, we use torchvision implementations (Paszke et al., 2019). We run the experiments on a single NVIDIA A100 GPU (80 GB, CUDA 12.2).

The memory usage for weights, gradients, and optimizer states can be easily computed by summing the sizes of all tensors of these kinds. To quantify the size of stored feature maps, we measure the allocated memory after the forward pass before the backward pass starts, and subtract the sizes of weights and optimizer states from the value. For runtime variables, we first identify the critical layer $l^*$ with the highest peak memory. In other words, the network-level peak memory occurs during backpropagation on layer $l^*$. The difference between the peak allocated memory at layer $l^*$ and the allocated memory at the start of the layer's backward pass reflects runtime variable costs. Note that gradient sizes are slightly overestimated, as not all gradients are ready when global peak memory is reached. Detailed results are shown in Table 6.

Table 6: Detailed memory breakdown of different networks when trained on ImageNet. Memory costs are measured in MB.

| Network | Stored Features | Weights | Gradients | Optimizer States | Runtime Variables |
|---------|-----------------|---------|-----------|------------------|-------------------|
| ResNet-34 | 936.49 | 83.15 | 83.15 | 83.15 | 38.25 |
| ViT | 1629.20 | 100.05 | 100.05 | 200.11 | 91.19 |
| SEW ResNet-34 | 8373.98 | 83.15 | 83.15 | 83.15 | 40.50 |
| Spikformer | 33372.02 | 113.26 | 113.26 | 226.52 | 496.00 |

## B    MEMORY EVOLUTION CURVES

Figure 3 and Figure 5 demonstrate the memory cost evolution within one training iteration of Spiking VGG on CIFAR10-DVS. To get these curves, we record the allocated memory at the start, peak, and end of each target layer's forward pass (FP) and backward pass (BP). The resulting sequence, arranged in the temporal order of events, is

$$\left[ \mathcal{M}_{\text{FP}^1}^{\text{start}}, \mathcal{M}_{\text{FP}^1}^{\text{peak}}, \mathcal{M}_{\text{FP}^1}^{\text{end}}, \mathcal{M}_{\text{FP}^2}^{\text{start}}, \mathcal{M}_{\text{FP}^2}^{\text{peak}}, \mathcal{M}_{\text{FP}^2}^{\text{end}}, \dots, \right.$$

$$\left. \mathcal{M}_{\text{BP}^2}^{\text{start}}, \mathcal{M}_{\text{BP}^2}^{\text{peak}}, \mathcal{M}_{\text{BP}^2}^{\text{end}}, \mathcal{M}_{\text{BP}^1}^{\text{start}}, \mathcal{M}_{\text{BP}^1}^{\text{peak}}, \mathcal{M}_{\text{BP}^1}^{\text{end}} \right], \tag{6}$$

where $\text{FP}^l$ and $\text{BP}^l$ denote the forward and backward pass of layer $l$, respectively. The global peak memory can be defined as $\mathcal{M}^{\text{peak}} = \max \left( \{\mathcal{M}_{\text{FP}^l}^{\text{peak}}\}_l \bigcup \{\mathcal{M}_{\text{BP}^l}^{\text{peak}}\}_l \right)$.

## C    DETAILS OF THE MAIN EXPERIMENTS

The main experiment is implemented using PyTorch (Paszke et al., 2019) and SpikingJelly (Fang et al., 2023a).

**Sequential CIFAR-10**    Sequential CIFAR-10 (Fang et al., 2023b; Chen et al., 2024) is a sequence classification task derived from the standard CIFAR-10 benchmark (Krizhevsky, 2009). It is widely used for evaluating SNNs' capability to learn long-term temporal patterns. In this task, the CIFAR-10 images are fed into the model column by column, mimicking the way humans scan pictures from left to right. Each sample is a sequence with $T = 32$ elements, and each element contains 32 RGB pixels.

There are $50,000$ training samples, $10,000$ test samples, and 10 classes. Following the practice in PSN (Fang et al., 2023b), we augment the training data with random mixup (Zhang et al., 2018), random cutmix (Yun et al., 2019), random horizontal flipping, TrivialAugment (Müller & Hutter, 2021), predefined data normalization, and random erasing. An 8-layer 1D convolutional SNN is employed (SCNN) (Fang et al., 2023b). Hyperparameters and running environment are listed in Table 7.

**DVS128 Gesture**  DVS128 Gesture (Amir et al., 2017) is an event-based gesture recognition dataset recorded by a DVS128 camera. It contains 11 gesture classes performed by 29 subjects under 3 illumination conditions with spatial resolution $128 \times 128$. For experiments, we follow the standard split provided in SpikingJelly (Fang et al., 2023a): 1,176 training samples and 288 test samples. Each recording is integrated into $T = 16$ frames, and no extra augmentations are applied. We use 7B-Net, a small-scale SEW ResNet (Fang et al., 2021a), as the backbone. See Table 7 for other hyperparameters and the running environment.

**CIFAR10-DVS**  CIFAR10-DVS (Li et al., 2017) is a neuromorphic vision classification task created by recording CIFAR-10 images (Krizhevsky, 2009) through a Dynamic Vision Sensor (DVS) (Lichtsteiner et al., 2008). The dataset is composed of $10,000$ samples, each represented as an event stream with 2 channels and $128 \times 128$ resolution. Following the protocol of temporal effective batch normalization (TEBN) (Duan et al., 2022) and PSN (Fang et al., 2023b), we partition the dataset into $9,000$ training samples and $1,000$ test samples, downsample the resolution to $48 \times 48$, and integrate each event stream into $T = 10$ frames. The data augmentation pipeline incorporates random resized cropping, random horizontal flipping, and Neuromorphic Data Augmentation (NDA) (Li et al., 2022). We adopt a Spiking VGG11 architecture, following the practice of TEBN (Duan et al., 2022) and PSN (Fang et al., 2023b). Refer to Table 7 for hyperparameters and running environment.

**ImageNet**  ImageNet-1k (Deng et al., 2009) is a large-scale visual recognition benchmark containing about 1.28 million training samples and 50,000 validation samples across 1,000 classes. Training on the entire ImageNet dataset is computationally expensive, so we use its $\frac{1}{32}$ subset instead, whose samples are evenly distributed across all 1000 classes. Since the peak memory cost during training is independent of the sample size, the memory footprint we report can faithfully reflect full-dataset training conditions. Each image is resized to $224 \times 224$ resolution. We utilize SEW ResNet-34 (Fang et al., 2021a), Spikformer (Zhou et al., 2023) and QKFormer (Zhou et al., 2024) architectures. The SEW residual connections in these architectures bring non-binary integer activation values (Fang et al., 2021a); for these activations, we compress them into 8-bit unsigned integers (uint8) rather than bits to avoid accuracy loss. For experiments using SEW ResNet-34, we use the same data augmentation pipeline as in the original work (Fang et al., 2021a); for both Spikformer and QKFormer, we augment data using the procedure in the original QKFormer work (Zhou et al., 2024). Hyperparameters and running environments are provided in Table 7.

Table 7: Hyperparameter settings and running environment configurations for the main experiments.

| | Sequential CIFAR-10 | DVS128 Gesture | CIFAR10-DVS | ImageNet | |
| --- | --- | --- | --- | --- | --- |
| | | | | SEW | Transformer |
| $\lambda$ | 0.5 | 0.5 | 0.25 | 0.5 | 0.5 |
| $V_{\text{th}}$ | 1.0 | 1.0 | 1.0 | 1.0 | 1.0 |
| Optimizer | SGD(0.9) | SGD(0.9) | SGD(0.9) | SGD(0.9) | AdamW |
| L2 Reg. | 0 | 0 | $5 \times 10^{-4}$ | 0 | $5 \times 10^{-2}$ |
| Init. LR | 0.1 | 0.1 | 0.1 | 0.1 | 0.001 |
| Scheduler | Cosine | Step(0.1, 64) | Cosine | Cosine | Cosine |
| Loss | CE | CE | TET | TET | Smooth CE |
| Batch Size | 128 | 16 | 32 | 32 | 32 |
| $T$ | 32 | 16 | 10 | 4 | 4 |
| $k$ | 2 | 2 | 2 | 2 | 2 |
| CUDA Version | 12.3 | 12.3 | 12.3 | 12.2 | 12.2 |
| Device | $1 \times$ 4090 | $1 \times$ 4090 | $1 \times$ 4090 | $1 \times$ A100 | $1 \times$ A100 |

## D    EXPERIMENTS OF MULTI-GPU TRAINING

The experiments in Table 1 of the main text are conducted on a single GPU. To further validate the scalability of the proposed framework, we conduct multi-GPU training experiments on ImageNet (Deng et al., 2009) using QKFormer (Zhou et al., 2024). The experimental setup follows Appendix C, except that 1, 2, or 3 NVIDIA A100 GPUs are used for distributed data parallel (DDP) training. We set a per-device batch size of 32. Table 8 reports the time and memory costs. Here, the batch time cost refers to the average time per training iteration for a single GPU. Throughput accounts for all GPUs, measured as the total number of training samples processed per second. The peak allocated memory is the maximum of peak allocated memory across all devices. Generally, in multi-GPU settings, our method achieves substantial memory efficiency improvements while incurring a moderate increase in training time, which is consistent with the single-GPU cases.

Table 8: Time and memory efficiency when training a QKFormer on ImageNet using multiple GPUs.

| #GPUs | Neuron | Method | Throughput (samples / s) $\uparrow$ | Peak. Alloc. Mem. (MB) $\downarrow$ |
|---|---|---|---|---|
| 1 | SJLIF | BPTT | 86.15 | 44571.33 |
|  | MELIF | $O_4$ | 76.51 $(0.89\times)$ | 5219.93 $(0.12\times)$ |
| 2 | SJLIF | BPTT | 168.43 | 44679.28 |
|  | MELIF | $O_4$ | 151.54 $(0.90\times)$ | 5323.13 $(0.12\times)$ |
| 3 | SJLIF | BPTT | 235.45 | 44679.28 |
|  | MELIF | $O_4$ | 211.01 $(0.90\times)$ | 5323.13 $(0.12\times)$ |

## E    ADDITIONAL MEMORY OPTIMIZATION TECHNIQUES

**Low-Memory Optimizer (LOMO)**    Low-memory optimization (LOMO) (Lv et al., 2024b) reduces the memory cost of gradients by updating $\mathbf{W}^l$ once its gradient $\mathbf{G}^l$ is computed, instead of waiting until all gradients are available. Unlike the stateless original LOMO (Lv et al., 2024b), we retain optimizer states (e.g., those of Adam (Kingma & Ba, 2015)) to match the baseline cases. LOMO ensures that at most one gradient tensor resides in memory at a time, reducing $\sum_l \mathcal{M}_{\mathbf{G}^l}$ in Equation (4) to $\max_l \mathcal{M}_{\mathbf{G}^l}$.

**Automatic Mixed Precision (AMP)**    Automatic mixed precision (AMP) training (Micikevicius et al., 2018) can be optionally enabled to reduce overall memory usage and accelerate training by utilizing 16-bit floats for activations and gradients. The loss is scaled to prevent underflow and ensure numerical stability.

## F    ACCURACY RESULTS

We report additional validation accuracy results in Table 9. Note that these experiments are designed to validate the mathematical equivalence of our method with standard BPTT rather than to maximize performance, so we do not apply advanced training tricks like random temporal delete (Fang et al., 2021a). We train 300, 192, and 100 epochs for Sequential CIFAR-10, DVS128 Gesture and CIFAR10-DVS, respectively. For $\frac{1}{32}$ ImageNet, we report validation accuracy at the fifth epoch to reduce training cost, which is sufficient to demonstrate the equivalence. The results show that MELIF attains accuracy nearly identical to SJLIF across all benchmarks, with minor discrepancies arising only from different backend numerical behaviors. In most cases, the optimization pipeline itself does not affect accuracy. However, $O_3$ and $O_4$ for Spikformer and QKFormer slightly influence accuracy due to the temporal segment partitioning on weight layers. Appendix G discusses this issue in detail. **These small deviations stem purely from numerical computation rather than from any approximation in the gradient computation. The gradients produced by our method remain free from systematic bias.**

Table 9: Comparison of validation accuracy (%). For $\frac{1}{32}$ ImageNet, we report the validation accuracy at epoch 5. For SJLIF conditions, we report mean $\pm$ std over three runs. For MELIF conditions, we report the results on a single run using a fixed seed.

| Task | Network | SJLIF | MELIF | | | | |
|---|---|---|---|---|---|---|---|
| | | BPTT | BPTT | $O_1$ | $O_2$ | $O_3$ | $O_4$ |
| Sequential CIFAR-10 | SCNN | 82.53±0.25 | 82.36 | 82.36 | 82.36 | 82.36 | 82.36 |
| DVS128 Gesture | 7B-Net | 95.08±0.87 | 95.14 | 95.14 | 95.14 | 95.14 | 95.14 |
| CIFAR10-DVS | Spiking VGG | 85.98±0.25 | 86.10 | 86.10 | 86.10 | 86.10 | 86.10 |
| $\frac{1}{32}$ ImageNet | SEW ResNet-34 | 3.46±0.21 | 3.50 | 3.50 | 3.50 | 3.50 | 3.50 |
| | Spikformer | 1.03±0.16 | 0.90 | 0.90 | 0.90 | 1.10 | 1.10 |
| | QKFormer | 1.06±0.12 | 1.20 | 1.20 | 1.20 | 1.10 | 1.10 |

## G  POTENTIAL SOURCES OF NUMERICAL DISCREPANCIES

**Numerical discrepancies in gradients may arise when temporal GC segment partitioning is applied to layers with learnable parameters.** Without temporal partitioning, the gradient is first computed for each time step $t \in \{1, \ldots, T\}$ and batch sample $n \in \{1, \ldots, N\}$, and then summed over the temporal and batch dimensions:

$$\mathbf{G} = \sum_{t=1}^{T} \sum_{n=1}^{N} \mathbf{G}_{t,n}, \tag{7}$$

where $\mathbf{G}_{t,n}$ denotes the gradient contribution at time step $t$ from sample $n$. In contrast, when the temporal dimension is partitioned ($k = 2$ for example), the accumulation is performed in two stages:

$$\mathbf{G}^{(1)} = \sum_{t=1}^{\frac{T}{2}} \sum_{n=1}^{N} \mathbf{G}_{t,n}, \quad \mathbf{G}^{(2)} = \sum_{t=\frac{T}{2}+1}^{T} \sum_{n=1}^{N} \mathbf{G}_{t,n}, \tag{8}$$

followed by a final aggregation:

$$\mathbf{G} = \mathbf{G}^{(1)} + \mathbf{G}^{(2)}. \tag{9}$$

Although mathematically equivalent to the unpartitioned case, these operations differ in numerical practice because **floating-point addition is not associative**. As a result, reordering the accumulation of gradient terms leads to slight deviations in the final gradient values. This explains the minor accuracy deviations observed in Table 9 for Spikformer and QKFormer at $O_3$ and $O_4$.

## H  TIME COST OF MEMORY OPTIMIZATION

Table 10 reports the time cost of the memory optimization pipeline at each optimization level. For SCNN, the overhead increases moderately with higher optimization levels, reflecting the additional computations from spatial and temporal segment partitioning and greedy segment restoration. For QKFormer, the jump in time cost from $O_3$ to $O_4$ is much more pronounced, primarily due to the transformer's greater depth, which increases profiling costs and the number of segments to iterate over. Importantly, this overhead is incurred only once before training and is negligible relative to the total training time.

Table 10: Time (in seconds) spent by the memory optimization pipeline at each optimization level.

| | $O_1$ | $O_2$ | $O_3$ | $O_4$ |
|---|---|---|---|---|
| SCNN | 1.08 | 26.32 | 30.63 | 75.41 |
| QKFormer | 1.13 | 44.91 | 78.58 | 564.26 |

## I  Tutorial

We provide a brief tutorial on using the proposed automatic memory optimization pipeline, taking the training of Spiking VGG on CIFAR10-DVS as an example. The model can be defined using PyTorch (Paszke et al., 2019) and SpikingJelly (Fang et al., 2023a) as shown in the code below.

```python
class VGGBlock(nn.Module):
    def __init__(
        self, in_plane, out_plane, T,
        neuron_type, preceding_avg_pool=False, **kwargs
    ):
        super().__init__()
        proj_bn = []
        if preceding_avg_pool:
            proj_bn.append(nn.AvgPool2d(2))
        proj_bn += [
            nn.Conv2d(in_plane, out_plane, 3, 1, 1),
            nn.BatchNorm2d(out_plane),
        ]
        self.proj_bn = SeqToANNContainer(*proj_bn)
        kwargs["T"] = T
        self.neuron = get_neuron(neuron_type, **kwargs)

    def forward(self, x_seq):
        return self.neuron(self.proj_bn(x_seq))

class CIFAR10DVSVGG(nn.Module):
    def __init__(self, T, neuron_type, dropout=0.25, **kwargs):
        super().__init__()
        self.features = nn.Sequential(
            VGGBlock(2, 64, T, neuron_type, False, **kwargs),
            VGGBlock(64, 128, T, neuron_type, False, **kwargs),
            VGGBlock(128, 256, T, neuron_type, True, **kwargs),
            VGGBlock(256, 256, T, neuron_type, False, **kwargs),
            VGGBlock(256, 512, T, neuron_type, True, **kwargs),
            VGGBlock(512, 512, T, neuron_type, False, **kwargs),
            VGGBlock(512, 512, T, neuron_type, True, **kwargs),
            VGGBlock(512, 512, T, neuron_type, False, **kwargs),
            layer.AvgPool2d(2, step_mode="m"),
        )
        d = int(48 / 2 / 2 / 2 / 2)
        l = [nn.Dropout(dropout)] if dropout > 0 else []
        l.append(nn.Linear(512 * d * d, 10))
        self.classifier = nn.Sequential(*l)
        for m in self.modules():
            if isinstance(m, nn.Conv2d):
                nn.init.kaiming_normal_(
                    m.weight, mode='fan_out', nonlinearity='relu'
                )

    def forward(self, input):
        # input.shape = [N, T, C, H, W]
        input = input.transpose(0, 1).contiguous()
        # [T, N, C, H, W]
        x = self.features(input)
        x = torch.flatten(x, 2)  # [T, N, D]
        x = self.classifier(x)
        return x
```

Users can define spatial partitioning rules by implementing the \_\_spatial\_split\_\_ method, which returns a tuple of submodules corresponding to the spatial subsegments of a layer. For instance, a VGG block can be split into a convolution-plus-batch-norm segment and a spiking neuron segment.

```python
class VGGBlock(nn.Module):
    def __spatial_split__(self):
        return self.proj_bn, self.neuron
```

To define temporal partitioning rules, users should implement the \_\_tc\_init\_states\_\_ and \_\_tc\_forward\_\_ methods. \_\_tc\_init\_states\_\_ returns a list of initial hidden states, while \_\_tc\_forward\_\_ takes a chunk of input tensors along with the initial hidden states, and then returns the corresponding outputs and updated hidden states. The stateless layer container SeqToANNContainer is the simplest case, where no hidden states are required and the temporally chunked forward pass is just the same as the container's original forward pass.

```python
class SeqToANNContainer(layer.SeqToANNContainer):
  """Stateless layer container that supports temporal chunking"""
  def __tc_init_states__(self, x_seq):
      return []

  def __tc_forward__(self, xc):
      return [self.forward(xc),]
```

A more complex example is the NeuronMaxPool block:

```python
class NeuronMaxPool(nn.Module):
    def __init__(self, neuron_type, **kwargs):
        super().__init__()
        self.neuron = get_neuron(neuron_type, **kwargs)
        self.pool = SeqToANNContainer(
            nn.MaxPool2d(kernel_size=3, stride=2, padding=1)
        )

    def forward(self, x_seq):
        return self.pool(self.neuron(x_seq))

    def __tc_init_states__(self, x_seq):
        device, dtype = x_seq.device, x_seq.dtype
        return [torch.zeros([], device=device, dtype=dtype)]

    def __tc_forward__(self, xc, v):
        sc, v = self.neuron.multistep_state_update(xc, v)
        yc = self.pool(sc)
        return yc, v
```

which means that the hidden state (the neuron's membrane potential) is initialized to zero, and the temporally forward pass consists of a multi-step state update of the neuron followed by max pooling. In this example, there is only one input, one hidden state, and one output. However, multiple inputs, hidden states, and outputs are also supported. Finally, the memory\_optimization function can be called to apply the automatic pipeline.

```python
net = CIFAR10DVSVGG(T, neuron_type, dropout, **kwargs)
net = memory_optimization(
    net,
    instance=(VGGBlock,),
    dummy_input=torch.rand(32, T, 2, 48, 48),
```

```
6        compress_x=True,
7        level=4,
8        verbose=True,
9        temporal_split_factor=2,
10   )
```

where `instance` specifies the layer types to apply gradient checkpointing, `dummy_input` is a sample input tensor for profiling, `compress_x` indicates whether to compress input spikes, `level` sets the optimization level, and `temporal_split_factor` is the $k$ factor that controls the granularity of temporal partitioning. After optimization, the model can be trained using standard procedures without further modification.

Note that the pipeline will automatically check whether spike compression is applicable at each GC segment based on the input distribution. Users can also manually specify spike compressors by setting the module's `x_compressor` attribute. For instance, for a layer in a SEW residual block (Fang et al., 2021a) whose input is non-binary integer tensors, we can compress the input to 8-bit unsigned integers (uint8):

```
1    class SEWBlock(nn.Module)
2        def __init__(self, c_in, c_mid, neuron_type, **kwargs):
3            super().__init__()
4            self.conv = nn.Sequential(
5                Conv3x3(c_in, c_mid, neuron_type, **kwargs),
6                Conv3x3(c_mid, c_in, neuron_type, **kwargs),
7            )
8            self.conv[0].x_compressor = "Uint8SpikeCompressor"
9
10       def forward(self, x: torch.Tensor):
11           out = self.conv(x)
12           out = out + x
13           return out
```

## J  THE EFFECT OF SPIKE COMPRESSION ON TRAINING MEMORY

Table 11 reports the peak memory usage corresponding to Figure 5. The majority of memory saving comes from layer-wise GC (BPTT vs. $O_1$, compression disabled), while spike compression alone only provides marginal memory savings ($O_1$, compression disabled vs. $O_1$, compression enabled). However, as shown in Figure 5, spike compression reduces the memory footprint of activations, thus substantially lowering the instantaneous memory usage of deeper layers. This reduction creates the headroom for stronger spatio-temporal partitioning, leading to larger memory savings at higher optimization levels ($O_3$ and $O_4$). In summary, **spike compression is not the main source of memory efficiency, but an enabling factor that allows spatio-temporal partitioning to further reduce peak memory.**

Table 11: Peak allocated memory (MB) of Spiking VGG when training on CIFAR10-DVS with spike compression enabled or disabled ($T = 10$, batch size is 32). See Figure 5.

| Compression | BPTT | $O_1$ (+GC) | $O_3$ (+partitioning) | $O_4$ (+restoration) |
|---|---|---|---|---|
| ✓ | / | 2887.75 | 2349.39 | 2349.39 |
| ✗ | 6131.07 | 2892.63 | 2530.66 | 2530.66 |

## K  DETAILED RUNTIME PROFILING

Table 12 reports the the forward and backward runtime for each layer in Spiking VGG. Note that the fully connected classification head is omitted since no change is applied to it across all optimization

levels. GC introduces additional backward computation roughly equivalent to a single extra local forward pass. Spike compression and decompression further add small overheads to both forward and backward passes, but the increase is negligible relative to the total runtime, demonstrating the efficiency of bit compression; note that Conv0 do not apply input spike compression. In this example, spatial partitioning is applied only to Conv1, while temporal partitioning is skipped since it does not yield additional memory benefits (see Algorithm 2). As a result, virtually no extra computational cost. Finally, greedy segment restoration significantly reduces the computation load of both passes. Conv3 and Conv5 are reverted to standard BPTT blocks, and their forward and backward runtimes return to BPTT level.

Table 12: Layer-wise runtime profiling for Spiking VGG on CIFAR10-DVS (MELIF, $T = 10$, batch size is 32). Results are averaged over 200 iterations with 10 warmup iterations and reported in milliseconds.

| Condition | Stage | Conv0 | Conv1 | Conv2 | Conv3 | Conv4 | Conv5 | Conv6 | Conv7 |
|---|---|---|---|---|---|---|---|---|---|
| BPTT | fwd | 2.48 | 5.96 | 4.16 | 5.22 | 2.80 | 3.98 | 1.04 | 0.91 |
| | bwd | 4.28 | 12.82 | 8.57 | 10.13 | 5.58 | 8.29 | 2.26 | 1.95 |
| + GC | fwd | 2.52 | 5.99 | 4.20 | 5.26 | 2.81 | 4.01 | 1.07 | 0.92 |
| | bwd | 6.80 | 18.86 | 12.78 | 15.42 | 8.45 | 12.33 | 3.27 | 2.89 |
| $O_1$ | fwd | 2.51 | 6.19 | 4.51 | 5.46 | 3.04 | 4.11 | 1.18 | 1.02 |
| | bwd | 6.81 | 19.08 | 13.21 | 15.63 | 8.71 | 12.46 | 3.41 | 3.04 |
| $O_3$ | fwd | 2.49 | **6.15** | 4.48 | 5.38 | 3.05 | 4.09 | 1.17 | 0.99 |
| | bwd | 6.72 | **19.02** | 13.24 | 15.62 | 8.56 | 12.22 | 3.39 | 2.98 |
| $O_4$ | fwd | 2.50 | 6.16 | 4.49 | **5.18** | 3.03 | **3.99** | 1.16 | 1.00 |
| | bwd | 6.76 | 19.01 | 13.23 | **10.08** | 8.55 | **8.24** | 3.34 | 2.97 |

We further investigate how training time cost scales with the number of checkpointed layers and temporal splits. Since GC performs one fixed-cost recomputation per segment, the total overhead increases as the number of GC segments grows. However, the scaling is not strictly linear, since recomputation cost varies across layers. As shown in the left plot of Figure 8, the overhead grows as more GC segments are added, but the increments are uneven. From Table 12, we know that spatial partitioning has almost no effect on training speed. In contrast, temporal partitioning actually reduces temporal parallelism, thereby slowing down training (especially for models with a large T). To illustrate this effect, we measure per-batch training time cost of DH-SFNN on SHD. Training time per batch increases nonlinearly with the the temporal partitioning factor $k$.

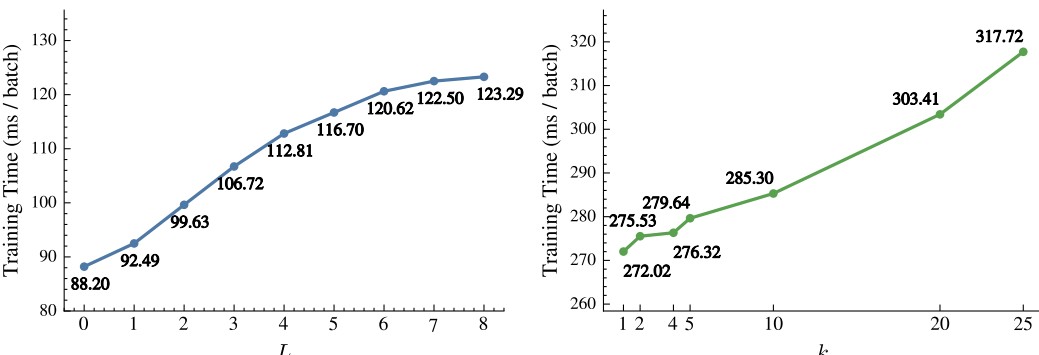

Figure 8: Left: taining time per batch (forward + backward) of Spiking VGG on CIFAR10-DVS as a function of the number of GC segments. The first $L$ layers adopt GC, where $L = 0$ corresponds to standard BPTT. $T = 10$, and batch size is 32. Right: training time per batch of DH-SFNN on SHD as a function of the temporal partitioning factor $k$ under $O_3$. $k = 1$ indicates no temporal partitioning. $T = 100$, and batch size is 128. Experiments are run on a single 4090 (24 GB).

## L   Fine-grained Ablation Study

Table 2 shows the ablation results of optimization levels on CIFAR10-DVS. To better understand the impact of each GC adjustment strategy, we conduct fine-grained ablations on QKFormer for ImageNet. As shown in Table 13, spatial partitioning provides substantial memory reduction with almost no impact on training throughput. Temporal partitioning also reduces memory usage, though it introduces a slight slowdown due to reduced temporal parallelism. When combined, the two strategies complement each other effectively, lowering peak memory to 5219 MB. Greedy restoration further improves throughput while preserving memory savings. With all three strategies jointly applied, throughput reaches 76.51 samples/s, the highest among all variants; notably, it even exceeds the condition without temporal partitioning because there are more GC segments restored to standard BPTT blocks. Overall, these ablations clarify the individual and collective contributions of the three components to memory and computational efficiency, confirming the intended synergistic effect of the full pipeline.

Table 13: Ablation study of QKFormer for ImageNet on the impact of spatial partitioning, temporal partitioning, and greedy restoration. All conditions adopt MELIF, GC and spike compression.

| Spatial Partition | Temporal Partition | Greedy Restoration | Throughput (sample/s) | Memory (MB) | Annotation |
|:---:|:---:|:---:|:---:|:---:|:---:|
| | | | 66.13 | 7726.55 | $O_1$ |
| ✓ | | | 66.02 | 6834.48 | $O_2$ |
| | ✓ | | 63.82 | 6920.25 | |
| | | ✓ | 73.17 | 7725.87 | |
| ✓ | ✓ | | 64.01 | 5219.93 | $O_3$ |
| ✓ | | ✓ | 73.07 | 6833.87 | |
| | ✓ | ✓ | 70.24 | 6920.25 | |
| ✓ | ✓ | ✓ | 76.51 | 5219.93 | $O_4$ |

## M   Lossless Spike Compressors

As discussed in Section 4.3, we adopt bit representation as the default lossless spike compressor due to its superior speed and memory efficiency. To validate this choice, we compare it with two alternatives: sparse representation (storing indices of non-zero elements) and lossless bit-stream compressor (e.g., ANS from nvCOMP [1]). Experiments are performed on Sequential CIFAR-10 using SCNN ($T = 32$, batch size is 128) on an NVIDIA GeForce RTX 4090. The results in Table 14 show that bit compression consistently achieves the lowest memory footprint and highest throughput under both $O_1$ and $O_4$. Sparse representation yields slightly higher memory consumption and lower speed, while ANS provides moderate compression gains but is substantially slower.

For a more direct comparison, we evaluate compressed size and compression-decompression time cost across a range of firing rates $\rho$, using a float32 spike tensor of $10^7$ elements (38.14 MB) as input. Time costs are averaged over 100 trials following 20 warm-up runs. As Table 15 and Table 16 show, bit compression consistently produces a fixed-size 1.19 MB representation regardless of sparsity, whereas sparse representation's memory efficiency decreases rapidly as $\rho$ increases. ANS achieves small compressed sizes at low sparsity but is over an order of magnitude slower. Notably, firing rates in modern activation-based SNNs typically fall within 0.02 to 0.35 (Zhou et al., 2024), a regime in which bit representation performs effectively. These results confirm that bit representation is both efficient and effective.

## N   Limitations, Future Work, and Social Impacts

While the proposed memory optimization pipeline achieves significant memory reduction with broad compatibility and preserved accuracy, several limitations remain. First, GC inevitably introduces computational overhead due to the recomputation of intermediate features during backward pass, and spike compression also slightly adds computational burden. Although we alleviate these by greedily

---

[1]https://developer.nvidia.com/nvcomp

Table 14: Comparison of lossless spike compressors (bit, sparse, ANS) on Sequential CIFAR-10 (SCNN, $T = 32$, batch size is 128).

| Compressor | Throughput (sample / s) | | Memory (MB) | |
|---|---|---|---|---|
| | $O_1$ | $O_4$ | $O_1$ | $O_4$ |
| bit | 496.79 | 474.98 | 4768.11 | 5138.76 |
| sparse | 527.53 | 516.99 | 4454.24 | 5020.27 |
| ANS | 509.42 | 497.71 | 2029.53 | 2542.50 |

Table 15: Compressed memory (MB) of spike compressors across firing rates $\rho$

| Compressor | $\rho = 0.01$ | $\rho = 0.1$ | $\rho = 0.2$ | $\rho = 0.5$ | $\rho = 0.8$ | $\rho = 0.9$ |
|---|---|---|---|---|---|---|
| bit | 1.19 | 1.19 | 1.19 | 1.19 | 1.19 | 1.19 |
| sparse | 0.76 | 7.63 | 15.27 | 38.14 | 61.03 | 68.66 |
| ANS | 0.30 | 1.68 | 2.82 | 5.14 | 6.61 | 6.95 |

Table 16: Compression time cost (sec) of spike compressors across firing rates $\rho$

| Compressor | $\rho = 0.01$ | $\rho = 0.1$ | $\rho = 0.2$ | $\rho = 0.5$ | $\rho = 0.8$ | $\rho = 0.9$ |
|---|---|---|---|---|---|---|
| bit | 0.1234 | 0.1257 | 0.1262 | 0.1216 | 0.1250 | 0.1215 |
| sparse | 0.1538 | 0.1568 | 0.1616 | 0.2210 | 0.2965 | 0.3119 |
| ANS | 2.6097 | 2.4416 | 2.4407 | 2.5761 | 2.6305 | 2.6603 |

restoring low-memory-impact GC segments to standard BPTT segments and implementing efficient Triton kernels for compression and decompression, the overall training speed can be reduced to about $0.8\times$ that of the baseline in the worst case. Second, our experiments mainly focus on visual and audio classification benchmarks, which is a common practice in SNN research. While the pipeline is theoretically applicable to other modalities, its effectiveness on tasks such as language modeling remains to be validated. Thirdly, the proposed method is specially designed for layer-wise training setting (Table 5), which is common for modern medium- to large-scale SNNs. Its applicability to strict online learning settings, where model parameters are updated immediately at each time step, is limited. Fourth, the current pipeline follows user-specified partitioning schemes to reduce the search space. Automatic generation of partitioning rules are not yet supported.

Future work can address these limitations in several directions. One direction is to evaluate the framework on large-scale SNNs for language tasks, and to design optimization strategies tailored to language backbones. This would broaden the applicability and further demonstrate the generalizability of the pipeline. Another promising direction is to automatically generate spatio-temporal partitioning rules, enabling fully automated memory optimization without manual intervention. To this end, more principled optimization approaches, such as dynamic programming (Gruslys et al., 2016), could be employed to search for efficient partitioning schemes.

By reducing the memory cost of SNN training while retaining BPTT-level accuracy and broad compatibility, our method lowers the hardware barriers for scaling up SNNs. The pipeline facilitates the deployment of energy-efficient SNNs on resource-constrained platforms, including mobile and edge IoT devices. Such advances can democratize access to neuromorphic computing, promote sustainable AI solutions, and ultimately contribute to reduced energy consumption in intelligent systems. We do not see any negative societal impacts from this work.

## O    USE OF LARGE LANGUAGE MODELS

We utilized large language models (LLMs) to refine phrasing, correct spelling and grammar, and enhance the clarity of expressions. Additionally, LLMs were employed to assist in result visualization, such as providing initial code templates or optimizing figure layout suggestions. However, the core ideas, methodological design, code framework development, and key contributions of this paper were independently conceived and completed by the authors, without relying on LLMs for substantive support.

