# OpenReview forum: "Towards Lossless Memory-efficient Training of Spiking Neural Networks via Gradient Checkpointing and Spike Compression"
_ICLR.cc/2026/Conference — ICLR 2026 Poster_

### Official Review · Reviewer_XHsK · 2025-10-26

**Soundness:** 3
**Presentation:** 3
**Contribution:** 2
**Rating:** 6
**Confidence:** 4

**Summary:**

The paper proposes to use gradient checkpoint (GC), possibly combined with compression, to reduce the peak memory required for training of SNN. The advantage of the proposed methods is that they do not introduce mathematical discrepancies (except for possible numerical ones), thus maintaining the training accuracy. The paper introduces a simple yet effective heuristic to minimize the peak memory by combining spatial and temporal GC segment partitioning. The paper is well written and easy to read.

Overall, the training performance penalty remains very limited (>0.9x on average) compared to the gain in peak memory (<0.4x on average), making the proposed method relevant and applicable to real world models training. However, the novelty and innovation of the paper remains limited: GC is a well-known technic, as well as compression.

**Strengths:**

- The paper is well written and easy to read;
- The peak memory reduction technic and heuristic proposed in the paper is relatively generic and applicable to many SNN models;
- The is a mathematical equivalency with the original model (despite possible numerical discrepancies);
- The impact on learning performance is very limited, compared to the gain on peak memory.

**Weaknesses:**

- The scope and impact of the paper remains limited;
- The novelty limited: GC optimization and compression are well-known technics;
- It is not clear how the proposed method will benefit the community: while some code snippets are provided in the supplementary materials, the possible release or diffusion of the source code is not mentioned.

**Questions:**

- Do the authors intent to release their method source code?
- It could be interesting for the authors to bring more insight about the different compression methods explored? In particular, by providing some figures on how they compare in terms of performances and peak memory?

---

> ### Author Response · Authors · 2025-11-19
> **To Reviewer XHsK**
>
> Thanks for your insightful feedback. The point-to-point responses are as follows.
>
> > **Weakness 1.** The scope and impact of the paper remain limited;
>
> While our work focuses on memory optimization for SNN training, we believe its significance is broad.
>
> * **Enabling larger and deeper SNNs.** Modern SNNs are rapidly scaling. From fully connected networks [1] (0.6M parameters, denoted as 1$\times$) in early ages to deep architectures like SEW ResNet [2]  (21.8M parameters for SEW ResNet34, 36.3$\times$) and Spikformer [3] (29.7M parameters, 49.5$\times$), and even spiking foundation models (680M parameters for mBART SpikeLM [4], 1133.3$\times$), the trend toward larger SNNs mirrors the evolution of ANNs. Many researchers argue that neuromorphic computing should advance towards a larger scale [5], with SNNs emerging as a promising candidate for this goal. Without effective training memory optimization, hardware limitations will constrain the scalability of SNNs.
> * **Enabling more time steps**. The additional time dimension of an SNN incurs significantly higher memory demands compared to a structurally similar ANN. Moreover, SNNs are typically trained synchronously on GPUs [6], while their inference on neuromorphic chips is asynchronous [7]. To narrow the gap between training and deployment, increasing the number of time steps $T$ is the most intuitive approach. Yet, it exacerbates memory burden during training. Memory optimization techniques like ours are essential for aligning SNN training with neuromorphic inference paradigms.
> * **Increasing accessibility.** While high-memory GPUs with 80GB of memory are commercially available, their prohibitively high costs render them inaccessible to many research groups. For example, a Google Scholar search for “2080 Ti” restricted to 2025 reveals over 1,000 papers still relying on this GPU with only 12GB of memory. By reducing memory overhead, our approach enables researchers with limited hardware resources to advance SNN innovation, democratizing access to this field.
> * **Active research direction.** In recent years, techniques like online learning and reversible networks have been proposed to mitigate memory bottlenecks in BPTT-based SNN training. These efforts reflect the community’s recognition of memory optimization as a priority, and our work aligns closely with this focus.
>
> In short, by addressing a critical and widely recognized bottleneck, our work extends the applicability of SNNs and provides practical impact for the research community of neuromorphic computing.
>
> > **Weakness 2.** The novelty limited: GC optimization and compression are well-known technics;
>
> We acknowledge that our work builds partly upon established techniques like GC and activation compression. However, our approach goes beyond a straightforward combination of these methods. Through fine-grained memory profiling, we identify the memory bottleneck of GC- and compression-augmented SNN training as residing in a small set of critical layers (Figure 3 of the manuscript). On this basis, we introduce **a novel multi-stage strategy to adaptively refine checkpoint placement**:
>
> * Spliting GC segment spatio-temporally to further reduce peak memory cost;
> * Greedily restoring GC segments to standard BPTT blocks for faster training without increasing peak memory.
>
> These strategies interact with spike compression (see our response to Reviewer pEn5's Question 1) and achieve high temporal and spatial efficiency, distinguishing our approach from standard GC or activation compression. In addition, we organize the optimization pipeline into **a clean and user-friendly interface** to facilitate adoption by researchers. Overall, our method provides a systematic and practical solution for memory-efficient training of modern deep SNNs, which has not been well addressed in prior work.
>
> > **Weakness 3.** It is not clear how the proposed method will benefit the community: while some code snippets are provided in the supplementary materials, the possible release or diffusion of the source code is not mentioned.
>
> > **Question 1.** Do the authors intend to release their method source code?
>
> We fully intend to **release the complete source code of our method upon acceptance**. We also plan to integrate the proposed memory optimization pipeline **into existing deep SNN programming frameworks** (e.g., SpikingJelly [6]) through pull requests to facilitate broader adoption by the community.

---

> ### Author Response · Authors · 2025-11-19
> **To Reviewer XHsK (Continued)**
>
> > **Question 2.** It could be interesting for the authors to bring more insight about the different compression methods explored? In particular, by providing some figures on how they compare in terms of performances and peak memory?
>
> As stated in Section 4.3 of the manuscript, we adopt **bit representation** as the default lossless spike compressor throughout the work due to its higher speed and memory efficiency. To justify the choice, we compare it with two alternatives mentioned in the manuscript: **sparse representation** (recording the indices of non-zero elements), and lossless bit-stream compressor (e.g., **ANS** provided by nvCOMP). Experiments are conducted on Sequential CIFAR-10 (SCNN, T=32, batch size=128, NVIDIA GeForce RTX 4090). The results show that bit compression achieves the lowest memory usage and highest throughput at both $O_1$ and $O_4$. Sparse representation yields slightly higher memory and lower speed, while ANS provides modest compression benefits but is substantially slower.
>
> |Compressor|Memory @ $O_1$ (MB)|Throughput @ $O_1$ (sample/s)|Memory @ $O_4$ (MB)|Throughput @ $O_4$ (sample/s)|
> |-|-|-|-|-|
> |bit|496.79|4768.11|474.98|5138.76|
> |sparse|527.53|4454.24|516.99|5020.27|
> |ANS|509.42|2029.53|497.71|2542.50|
>
> For a more straightforward comparison, we directly evaluate the compressed size and compression-decompression time cost across a range of firing rate $\rho$, using a float32 spike tensor of $10^7$ elements as input (38.14 MB memory). Time costs are averaged over 100 trials with 20 warm-up trials. Bit compression consistently produces a fixed-size 1.19 MB representation regardless of sparsity, while sparse representation's memory efficiency degrades rapidly as $\rho$ increases. ANS, although producing small compressed sizes at low sparsity, is more than an order of magnitude slower. Note that firing rate in modern activation-based SNNs is typically within the range 0.02 to 0.35 [8], where bit representation can generally work well.
>
> *Memory (MB):*
>
> |Compressor|$\rho=0.01$|$\rho=0.1$|$\rho=0.2$|$\rho=0.5$|$\rho=0.8$|$\rho=0.9$|
> |-|-|-|-|-|-|-|
> |bit|1.19|1.19|1.19|1.19|1.19|1.19|
> |sparse|0.76|7.63|15.27|38.14|61.03|68.66|
> |ANS|0.30|1.68|2.82|5.14|6.61|6.95|
>
> *Time cost (sec):*
>
> |Compressor|$\rho=0.01$|$\rho=0.1$|$\rho=0.2$|$\rho=0.5$|$\rho=0.8$|$\rho=0.9$|
> |-|-|-|-|-|-|-|
> |bit|0.1234|0.1257|0.1262|0.1216|0.1250|0.1215|
> |sparse|0.1538|0.1568|0.1616|0.2210|0.2965|0.3119|
> |ANS|2.6097|2.4416|2.4407|2.5761|2.6305|2.6603|
>
> These results confirm that bit representation is both efficient and effective. We will add these results to the revised Appendix.
>
> **References**
>
> [1] Wu, Yujie, et al. "Spatio-temporal backpropagation for training high-performance spiking neural networks." Frontiers in neuroscience 12 (2018): 331.
>
> [2] Fang, Wei, et al. "Deep residual learning in spiking neural networks." Advances in Neural Information Processing Systems 34 (2021): 21056-21069.
>
> [3] Zhou, Zhaokun, et al. "Spikformer: When Spiking Neural Network Meets Transformer." The Eleventh International Conference on Learning Representations (2023).
>
> [4] Xing, Xingrun, et al. "SpikeLM: Towards General Spike-Driven Language Modeling via Elastic Bi-Spiking Mechanisms." International Conference on Machine Learning. PMLR, 2024.
>
> [5] Kudithipudi, Dhireesha, et al. "Neuromorphic computing at scale." Nature 637.8047 (2025): 801-812.
>
> [6] Fang, Wei, et al. "Spikingjelly: An open-source machine learning infrastructure platform for spike-based intelligence." Science Advances 9.40 (2023): eadi1480.
>
> [7] Yao, Man, et al. "Spike-based dynamic computing with asynchronous sensing-computing neuromorphic chip." Nature Communications 15.1 (2024): 4464.
>
> [8] Zhou, Chenlin, et al. "Qkformer: Hierarchical spiking transformer using qk attention." Advances in Neural Information Processing Systems 37 (2024): 13074-13098.

---

### Official Review · Reviewer_eX6a · 2025-10-27

**Soundness:** 3
**Presentation:** 3
**Contribution:** 3
**Rating:** 6
**Confidence:** 3

**Summary:**

This paper presents an automatic, lossless memory optimization pipeline for training spiking neural networks (SNNs) using backpropagation through time (BPTT). The core idea is to combine layer-wise gradient checkpointing with lossless spike compression, targeting the two major memory bottlenecks in SNN training—internal state storage and per-layer spike activations.

**Strengths:**

Originality
1. Introduces a lossless and automatic memory-saving approach for SNNs, avoiding the accuracy compromises typical in online learning or reversible architectures.
2. The combination of gradient checkpointing with binary spike compression is novel and elegantly leverages SNN characteristics (binary activations).

Quality
1. Provides comprehensive theoretical analysis (Equations 2–5) for memory cost and correctness.
2. Extensive empirical validation on multiple architectures and datasets.
3. Clear comparisons with other efficiency-oriented methods (online learning, BPTT-to-BP, reversible networks).

Clarity
Well-structured paper with good logical flow. Figures illustrate the differences between BPTT, checkpointing, and the proposed compression. The language is clear and technically rigorous.

Significance
1. Addresses one of the most critical barriers in SNN research: high memory cost during training.
2. Enables scaling SNNs to large architectures and long sequences, potentially democratizing access to neuromorphic research on commodity GPUs.

**Weaknesses:**

1. Limited exploration of trade-offs: Although the paper claims ≤20% slowdown, more fine-grained runtime profiling across levels would strengthen the argument for scalability. The additional computational cost of spike compression/decompression could be analyzed more quantitatively.
2. Sparse ablation: The adaptive checkpoint adjustment (spatial vs. temporal) is key but not deeply evaluated in isolation; Ablations showing how each component (spatial partitioning, temporal partitioning, greedy restoration) contributes to performance would enhance interpretability.

**Questions:**

1. Checkpoint adaptation: How sensitive is the memory efficiency to the chosen “level” parameter (O1–O4)? Could adaptive tuning be integrated dynamically during training? How does the system decide spatial vs. temporal split thresholds? Could these be learned or auto-tuned?
2. The paper reports ≤20% slowdown, but could the authors provide more detailed runtime decomposition? Does the time overhead scale linearly with the number of checkpoints or show nonlinear interactions with spatial/temporal splits?

---

> ### Author Response · Authors · 2025-11-19
> **To Reviewer eX6a**
>
> Thanks for your valuable comments. The point-to-point responses are as follows.
>
> > **Weakness 1.** Limited exploration of trade-offs: Although the paper claims $\le$20% slowdown, more fine-grained runtime profiling across levels would strengthen the argument for scalability. The additional computational cost of spike compression/decompression could be analyzed more quantitatively.
>
> > **Question 2.** The paper reports $\le$20% slowdown, but could the authors provide more detailed runtime decomposition? Does the time overhead scale linearly with the number of checkpoints or show nonlinear interactions with spatial/temporal splits?
>
> We thank the reviewer for the helpful suggestions. Here, we conduct a detailed layer-wise runtime profiling for Spiking VGG on CIFAR10-DVS (MELIF, batch size=32, $T$=10, averaged over 200 iterations with 10 warmup iterations, reported in milliseconds).
>
> |Condition|Stage|Conv0|Conv1|Conv2|Conv3|Conv4|Conv5|Conv6|Conv7|FC|
> |-|-|-|-|-|-|-|-|-|-|-|
> |std. BPTT|forward|2.48|5.96|4.16|5.22|2.80|3.98|1.04|0.91|0.21|
> |std. BPTT|backward|4.28|12.82|8.57|10.13|5.58|8.29|2.26|1.95|0.20|
> |+ layer-wise GC|forward|2.52|5.99|4.20|5.26|2.81|4.01|1.07|0.92|0.20|
> |+ layer-wise GC|backward|6.80|18.86|12.78|15.42|8.45|12.33|3.27|2.89|0.19|
> |+ compression ($O_1$)|forward|2.51|6.19|4.51|5.46|3.04|4.11|1.18|1.02|0.21|
> |+ compression ($O_1$)|backward|6.81|19.08|13.21|15.63|8.71|12.46|3.41|3.04|0.19|
> |+ st. partition ($O_3$)|forward|2.49|**6.15**|4.48|5.38|3.05|4.09|1.17|0.99|0.19|
> |+ st. partition ($O_3$)|backward|6.72|**19.02**|13.24|15.62|8.56|12.22|3.39|2.98|0.18|
> |+ restoration ($O_4$)|forward|2.50|6.16|4.49|**5.18**|3.03|**3.99**|1.16|1.00|0.20|
> |+ restoration ($O_4$)|backward|6.76|19.01|13.23|**10.08**|8.55|**8.24**|3.34|2.97|0.18|
>
> Key observations are as follows:
>
> 1. GC increases backward time cost by an amount closely matching one additional local forward pass.
> 2. Spike compression and decompression slightly add computational cost to both forward and backward passes. This increase is negligible compared to the overall runtime, confirming the efficiency of bit compression. Note that Conv0 and FC do not apply input spike compression.
> 3. In this example, spatial partitioning is applied only to Conv1, while temporal partitioning is not applied since it produces no further memory efficiency gain (refer to Algorithm 2 in the manuscript). In this case, partitioning adds virtually no extra computational cost.
> 4. Greedy segment restoration substantially reduces forward and backward time costs. In this example, Conv3 and Conv5 are restored to standard BPTT blocks. Their forward and backward time costs are reverted to BPTT level.
>
> Regarding the reviewer's question on scaling behavior:
>
> * Because GC performs one fixed-cost recomputation per segment (Observation 1), the total overhead increases as the number of GC segments grows. However, the scaling is not strictly linear, since recomputation cost varies across layers. To verify this, we apply layer-wise GC to the first $L$ layers of Spiking VGG on CIFAR10-DVS and measure per-batch training time (forward + backward, batch size=32, $T$=10, averaged over 3 epochs). These results confirm that the overhead grows as more GC segments are added, but the increments are uneven.
>
> | $L$| 0 (std. BPTT) | 1     | 2     | 3      | 4      | 5      | 6      | 7      | 8 ($O_1$) |
> |-|-|-|-|-|-|-|-|-|-|
> | Training Time (ms / batch) | 88.20      | 92.49 | 99.63 | 106.72 | 112.81 | 116.70 | 120.62 | 122.50 | 123.29    |
>
> * As shown in Observation 3, spatial partitioning has almost no effect on training speed.
> * Though not visible in the Spiking VGG profiling, temporal partitioning actually reduces temporal parallelism, thereby slowing down training (especially for models with a large T). To illustrate this effect, we measure per-batch training time cost of DH-SFNN on SHD (batch size=128, $T$=100; see Section 5.5 of the manuscript). $O_3$ optimization is applied, with temporal partitioning factor $k$ set as different values. Training time increases nonlinearly with $k$.
>
> | $k$ | 1 (no partition) | 2     | 4     | 5     | 10    | 20     | 25     |
> |-|-|-|-|-|-|-|-|
> | Training Time (ms / batch) | 272.02     | 275.53 | 276.32 | 279.64 | 285.30 | 303.41 | 317.72 |
>
> We will include these profiling results and clarifications in the revised Appendix.

---

> ### Author Response · Authors · 2025-11-19
> **To Reviewer eX6a (Continued)**
>
> > **Weakness 2.** Sparse ablation: The adaptive checkpoint adjustment (spatial vs. temporal) is key but not deeply evaluated in isolation; Ablations showing how each component (spatial partitioning, temporal partitioning, greedy restoration) contributes to performance would enhance interpretability.
>
> We appreciate the reviewer’s suggestion to further isolate the contribution of each component. To address this concern, we conduct an additional ablation study on QKFormer for ImageNet. All conditions adopt MELIF, layer-wise GC and input spike compression, while spatial partitioning, temporal partitioning and greedy restoration are selectively enabled. For other settings, refer to Appendix C. The measured training throughput and peak allocated memory are summarized below.
>
> |Spatial Partition|Temporal Partition|Greedy Restoration|Throughput (sample / s)|Memory (MB)|Annotation|
> |-|-|-|-|-|-|
> ||| |66.13|7726.55|$O_1$|
> |✔︎|| |66.02|6834.48|$O_2$|
> ||✔︎| |63.82|6920.25||
> |||✔︎|73.17|7725.87||
> |✔︎|✔︎| |64.01|5219.93|$O_3$|
> |✔︎||✔︎|73.07|6833.87||
> ||✔︎|✔︎|70.24|6920.25||
> |✔︎|✔︎|✔︎|76.51|5219.93|$O_4$|
>
> Key observations are as follows:
>
> * Spatial partitioning significantly reduces memory usage with almost no impact on throughput.
> * Temporal partitioning also reduces memory, but there is a slight drop in throughput.
> * When combining temporal partitioning with spatial partitioning, peak memory is reduced to 5219 MB, indicating strong complementarity between the two partition strategies.
> * Greedy restoration consistently improves throughput without harming memory efficiency.
> * When all three strategies are jointly applied, throughput reaches 76.51 samples/s, the highest among all variants. It is even faster than the condition without temporal partitioning, because more redundant GC segments are correctly identified and restored.
>
> Overall, these ablations disentangle how each component contributes to memory reduction and computational efficiency, validating the intended synergistic effect of our full pipeline. We will include this full ablation table and analysis in the revised Appendix.
>
> > **Question 1.** Checkpoint adaptation: How sensitive is the memory efficiency to the chosen “level” parameter (O1–O4)? Could adaptive tuning be integrated dynamically during training? How does the system decide spatial vs. temporal split thresholds? Could these be learned or auto-tuned?
>
> We thank the reviewer for the thoughtful questions. We address each point below.
>
> **Sensitivity to optimization level**
>
> As an example, **Table 2** in the manuscript summarizes the influence of optimization level on throughput and peak allocated memory. $O_1$ (layer-wise GC + spike compression) is the main source of memory reduction, but it substantially slows down training. $O_2$ and $O_3$ (+ spatial and temporal GC segment partitioning) further reduce peak memory usage. $O_4$ (+ greedy segment restoration) significantly speeds up training while keeping the memory cost equal to that of $O_3$. Compared with $O_1$, $O_4$ improves throughput to about $1.10\times$ and reduces peak memory cost to about $0.81\times$.
>
> **Possibility of dynamic adaptation during training**
>
> Our memory optimization pipeline is performed once before training using a dummy input. Since the memory and runtime characteristics of each network component remain highly stable throughout training, the profiling results before training reliably reflect actual behavior if the dummy input is properly set. Thus, dynamic tuning during training is generally not necessary. However, in rare cases where the training environment changes significantly over time (e.g., GPU failures in multi-device setups, or resource contention when new processes appear on the devices), re-profiling becomes necessary. In such scenarios, users can simply re-invoking our `memory_optimization` pipeline periodically (e.g., every $N$ epochs).
>
> **How spatial and temporal split rules are determined**
>
> Currently, the spatial partitioning rules, temporal partitioning rules, and the temporal partitioning factor $k$ should be specified by user, as demonstrated in Appendix I. Users may search for a suitable $k$ value via lightweight sweeps (i.e., calling `memory_optimization` multiple times without actual training). Also, heuristic-based partition rules typically work well in practice. We plan to explore automatic rule generation and hyperparameter autotuning in future work.

---

### Official Review · Reviewer_pEn5 · 2025-10-30

**Soundness:** 3
**Presentation:** 3
**Contribution:** 3
**Rating:** 8
**Confidence:** 3

**Summary:**

Training spiking neural networks (SNNs) is highly memory-intensive, typically requiring O(LT) memory. This paper addresses the issue by taking advantage of the binary nature of spikes and adopting a checkpointing strategy. During training, the computational graph is not constructed; instead, the input data is passed through the network while intermediate activations, if they are spikes, are compressed to reduce memory usage. Once the loss at the final layer is computed, a local computational graph is reconstructed at each layer, starting from the final layer and proceeding backward to the first layer, using the stored intermediate representations as inputs. This strategy significantly reduces the memory footprint during training.

**Strengths:**

Paper is easy to follow.

The proposed method can be easily adapted to train SNN models in a memory-efficient way.

**Weaknesses:**

I guess it's not very efficient if the SNN models are trained online, here it is assumed that the entire spiking data is available.

**Questions:**

1. Is it correct that the main source of memory efficiency arises from the fact that spike activations can be compressed, rather than being stored as 32-bit floating-point representations?

2. This method appears to rely on having access to the entire sequence of T input spikes during training. In an online setting, where input spikes arrive sequentially, how would the model behave? Could the authors comment on its applicability and performance in such scenarios?

3. The checkpointing strategy has also been successfully applied to train Neural ODEs [1]. Given the conceptual similarities between SNNs and NODEs from the ODE perspective, the authors may consider citing this related work for completeness.

4. Could the authors provide results obtained after training the models for the full number of epochs (e.g., on DVS-CIFAR10 or other datasets)? In the paper, only partial training results are presented to demonstrate the memory advantage. It would be valuable to also show that the accuracy remains unaffected when the model is trained to convergence.

[1] https://proceedings.mlr.press/v119/zhuang20a/zhuang20a.pdf

---

> ### Author Response · Authors · 2025-11-19
> **To Reviewer pEn5**
>
> Thanks for your constructive comments and suggestions. Our responses to your concerns and questions are as follows.
>
> > **Weakness 1.** I guess it's not very efficient if the SNN models are trained online, here it is assumed that the entire spiking data is available.
>
> > **Question 2.** This method appears to rely on having access to the entire sequence of T input spikes during training. In an online setting, where input spikes arrive sequentially, how would the model behave? Could the authors comment on its applicability and performance in such scenarios?
>
> We appreciate the reviewer’s question regarding step-wise or online training. Our method is specially designed for layer-wise training setting (Table 5). For step-wise training scenarios, two cases can be distinguished:
>
> 1. **Accumulating updates across time steps.** In this case, gradients from each time step are accumulated, and the parameters are updated only after the full sequence is processed. With careful implementation (e.g., reformulate the loss, and truncate temporal gradient flow), the computation can be rearranged into layer-wise style ($T$-step layer-wise forward $\to$ loss calculation $\to$ layer-wise backward) while preserving equivalence. Then, our method is applicable.
> 2. **Strict step-by-step (online) updates.** In this case, parameters are updated immediately at each time step, and there is no temporal gradient flow. Since the computation within each time step is effectively layer-wise, some components of our pipeline (layer-wise GC, spike compression, spatial partitioning, and greedy restoration) remain applicable to single-step computation, whereas temporal partitioning is not applicable.
>
> Despite partial applicability, this work pays less attention to step-wise or online setting because:
>
> 1. The memory footprint for online SNN training is very small, since only the computational graph of the current time step is stored (see Table 5 and Related Work in the manuscript). The room for further memory savings is limited.
> 2. Step-wise setting is not compatible with mainstream SNN training acceleration techniques like temporal parallelization and temporal kernel fusion, resulting in slower training than layer-wise setting [1].
> 3. Most recent medium- to large-scale SNNs are trained in layer-wise mode, not step-wise mode [2-4].
>
> For these reasons, we focus on layer-wise setting, where memory is a practical bottleneck and our method provides substantial benefits.
>
> > **Question 1.** Is it correct that the main source of memory efficiency arises from the fact that spike activations can be compressed, rather than being stored as 32-bit floating-point representations?
>
> We have demonstrated the effect of spike compression using memory evolution curves in Figure 5 of the manuscript. To further clarify this, we list the memory usage (MB) of Spiking VGG when trained on CIFAR10-DVS with spike compression enabled or disabled ($T$=10, batch size=32).
>
> | Compression | BPTT | $O_1$ (+GC) | $O_3$ (+partitioning) | $O_4$ (+restoration) |
> |-|-|-|-|-|
> | enabled | / | 2887.75 | 2349.39 | 2349.39 |
> | disabled | 6131.07 | 2892.63 | 2530.66 | 2530.66 |
>
> We can see that **the dominant source of memory saving comes from layer-wise GC** (BPTT vs. $O_1$, compression disabled), while spike compression alone only provides marginal memory savings ($O_1$, compression disabled vs. $O_1$, compression enabled). However, as shown in Figure 5 of the manuscript, spike compression reduces the memory footprint of activations, thus substantially lowering the instantaneous memory usage of deeper layers. This reduction creates the headroom for stronger spatio-temporal partitioning, leading to larger memory savings at higher optimization levels ($O_3$ and $O_4$). In other words, **spike compression is not the main source of memory efficiency, but an enabling factor that allows spatio-temporal partitioning to further reduce peak memory**. We will include this discussion in the revised manuscript.
>
> > **Question 3.** The checkpointing strategy has also been successfully applied to train Neural ODEs [Adaptive Checkpoint Adjoint Method for Gradient Estimation in Neural ODE, ICML2020]. Given the conceptual similarities between SNNs and NODEs from the ODE perspective, the authors may consider citing this related work for completeness.
>
> We thank the reviewer for pointing out this relevant work. The adaptive checkpointing strategy applied to iterative dynamical models proposed in the mentioned work is relevant to our focus. We will include a discussion of this paper in Section 3.2 of the revised manuscript to improve completeness.

---

> ### Author Response · Authors · 2025-11-19
> **To Reviewer pEn5 (Continued)**
>
> > **Question 4.** Could the authors provide results obtained after training the models for the full number of epochs (e.g., on DVS-CIFAR10 or other datasets)? In the paper, only partial training results are presented to demonstrate the memory advantage. It would be valuable to also show that the accuracy remains unaffected when the model is trained to convergence.
>
> The results after full training have already been included in **Appendix F (Table 9)**, where we report the final accuracies on Sequential CIFAR-10 (300 epochs), DVS Gesture (192 epochs), and CIFAR10-DVS (100 epochs). For the reviewer’s convenience, we present key results from Table 9 directly here. These experiments aim to verify the mathematical equivalence of the proposed method with standard BPTT rather than to maximize accuracy, so we do not employ advanced training tricks or fine-grained tuning. Minor accuracy differences arise only from the different numerical behavior of neuron backends (CuPy for SJLIF, and Triton for MELIF). The memory optimization pipeline itself does not affect final accuracies in these experiments even under $O_3$ or $O_4$, since temporal partitioning is not applied to the weight layers in these cases (see Appendix G for explanation).
>
> | Task | Network | SJLIF, BPTT | MELIF, BPTT | MELIF, $O_1$ | MELIF, $O_2$ | MELIF, $O_3$ | MELIF, $O_4$ |
> |-|-|-|-|-|-|-|-|
> | Sequential CIFAR-10 | SCNN | 82.53$\pm$0.25  | 82.36  | 82.36 | 82.36 | 82.36 | 82.36 |
> | DVS128 Gesture | 7B-Net | 95.08$\pm$0.87  | 95.14  | 95.14 | 95.14 | 95.14 | 95.14 |
> | CIFAR10-DVS | Spiking VGG | 85.98$\pm$0.25  | 86.10  | 86.10 | 86.10 | 86.10 | 86.10 |
>
> **References**
>
> [1] Fang, Wei, et al. "Spikingjelly: An open-source machine learning infrastructure platform for spike-based intelligence." Science Advances 9.40 (2023): eadi1480.
>
> [2] Zhou, Zhaokun, et al. "Spikformer: When Spiking Neural Network Meets Transformer." The Eleventh International Conference on Learning Representations (2023).
>
> [3] Xing, Xingrun, et al. "SpikeLM: Towards General Spike-Driven Language Modeling via Elastic Bi-Spiking Mechanisms." International Conference on Machine Learning. PMLR, 2024.
>
> [4] Zou, Shihao, et al. "SpikeVideoFormer: An Efficient Spike-Driven Video Transformer with Hamming Attention and $\mathcal {O}(T)$ Complexity." International Conference on Machine Learning. PMLR, 2025.

---

> > ### Comment · Reviewer_pEn5 · 2025-11-26
> > **Thanks for your response**
> >
> > Thanks for clarifying my doubts. I have already given a high score and will stick with that.

---

### Official Review · Reviewer_cB7e · 2025-10-30

[review text omitted: it was posted to a different submission]

---

> ### Author Response · Authors · 2025-11-12
> **Regarding Potential Misassignment of Review Comments**
>
> Dear Reviewer cB7e,
>
> We sincerely appreciate your time and effort in reviewing our manuscript. However, it appears that **the comments provided may have been mistakenly associated with our submission, as they seem to refer to a different paper**. We would like to kindly bring this to your attention in case there was an accidental mix-up during the review process. If our understanding is incorrect, we would be very grateful for your clarification and further guidance.
>
> Thank you again for your time and consideration.

---

### Author Response · Authors · 2025-11-25
**Manuscript Revision Summary**

Dear Reviewers,

Thank you for your valuable comments and suggestions. We have carefully revised the manuscript accordingly. Here is a summary of the key revisions:

* Add a discussion about spike compression's impact on memory cost and the dominant source of memory reduction to Appendix J.
* Include a discussion of [Adaptive Checkpoint Adjoint Method for Gradient Estimation in Neural ODE, ICML2020] to Section 3.2.
* Add a detailed runtime decomposition and an analysis of training time cost's scaling behavior to Appendix K.
* Include a fine-grained ablation study of the proposed GC adjustment strategies in Appendix L.
* Include a comparison of different spike compressors in Appendix M.
* Include a statement of LLM usage in Appendix O.
* Add an open-source statement to Abstract.

Revised contents in the PDF are highlighted in blue for easy reference. We sincerely hope these revisions address your concerns effectively. Please feel free to contact us if you require further clarification.

---

### Author Response · Authors · 2025-12-02
**Summary of Reviews and Responses**

We provide here a concise summary to facilitate the AC’s assessment. Note: **Reviewer cB7e's comments appear to correspond to a different submission**, and are therefore not included.

**Strengths**

* All reviewers acknowledged the **excellent memory efficiency** of our method.
* Reviewer pEn5 highlighted that the proposed method is **easy to use**.
* Reviewer eX6a praised the comprehensive **theoretical analysis** on memory cost, extensive **empirical validation** on multiple architectures and datasets, and clear **comparison** with other memory-efficient SNN training methods.
* Reviewer XHsK recognized the **wide applicability** and **high training performance** of the method.

**Issues and Supplementary Experiments**

* Reviewer pEn5 questioned whether spike compression is the **main source of memory saving**. We clarified by experiments that layer-wise gradient checkpointing is the dominant contributor to memory saving, while spike compression enables further reduction through spatio-temporal partitioning (see Appendix J).
* Reviewer eX6a pointed out the lack of **fine-grained runtime profiling and isolated ablation of core components (spatial/temporal partitioning, greedy restoration)**. We conducted detailed layer-wise runtime decomposition and analyzed the scaling behavior of time overhead in Appendix K. We also performed ablation studies to disentangle the contribution of each adjustment strategy in Appendix L.
* Reviewer XHsK requested more insights into **compression methods**. We provided systematic comparisons of different compressors (bit, sparse, ANS) showing that bit compression provides the best trade-off between speed and memory efficiency (see Appendix M).

**Controversies**

* Reviewer XHsK raised concerns about this work's scope and impact. However, Reviewer eX6a highlighted that **memory-efficient training is crucial** for SNNs' scaling to large models and long sequences, and can democratize access to neuromorphic research on commodity GPUs. We further elaborated on the broad impact (see our response to Reviewer XHsK).
* Reviewer XHsK questioned the novelty of the proposed method, as gradient checkpointing and compression are well-known techniques. However, Reviewer eX6a acknowledged the **novelty** of our combination of gradient checkpointing with binary spike compression, and commented that this combination elegantly leverages SNN characteristics. We also highlighted our originality in the multi-stage adaptive checkpoint refinement strategy (spatio-temporal partitioning + greedy restoration) and the unique design of a user-friendly interface.

**Additional Clarifications**

* Reviewer pEn5 questioned the method’s applicability in online training settings. We clarified that the method is designed for layer-wise training, which is the mainstream training paradigm for modern medium- or large-scale SNNs. We also point out that it is less necessary to optimize online training memory, since its memory cost is already very small.
* Reviewer pEn5 pointed out a relevant work on gradient checkpointing for neural ordinary differential equations. We acknowledged the relevance and included a discussion of it in our manuscript.
* Reviewer pEn5 requested the training results on full epochs. We argued that Table 9 in Appendix F of the original manuscript can address this issue.
* Reviewer eX6a questioned about the sensitivity of memory efficiency to optimization level, the possibility of dynamic tuning, and how spatio-temporal partitioning rules are decided. We explained the impact of each level by referring to Table 2 of the manuscript, and noted that dynamic tuning is unnecessary in most cases but feasible via periodic re-profiling. We also acknowledged that the partitioning rules are currently manually specified and left auto-partitioning to future work.
* Reviewer XHsK raised concerns about code release. We confirmed the plan to release the source code and integrated it into mainstream SNN frameworks.

Also, see **Manuscript Revision Summary** for a summary of the revisions made to the manuscript.

---

### Meta-Review · Area_Chair_Ssk2 · 2026-01-06

**Summary:**

This submission proposes an automatic, lossless memory-optimization pipeline for training spiking neural networks (SNNs) with BPTT, combining layer-wise gradient checkpointing and lossless spike compression, together with a multi-stage checkpoint refinement strategy (spatio-temporal segment partitioning + greedy restoration) guided by profiling. The core message is that the method achieves substantial peak-memory reduction (up to ~8×) while preserving BPTT-equivalent accuracy and maintaining a modest training speed impact in practice.

Overall, the technical evaluation is positive. Two reviewers (eX6a, XHsK) rate the paper at the borderline accept level (6) but recognize that the method is sound, broadly applicable, and practical. Another reviewer (pEn5) is strongly supportive (8) and explicitly confirms after discussion that the rebuttal addressed their concerns. One review (cB7e) is clearly misassigned and discusses LLM KV-cache/attention compression rather than SNN training; it should not be used for decision-making.

**Reviewer Concerns:**

Addressed Concerns:
- Practical significance and broad applicability: Memory is a primary bottleneck in BPTT-based SNN training; the proposed pipeline is presented as a generally applicable, user-facing optimization pass that can benefit many architectures/tasks.
- Lossless nature (no algorithmic approximation): Unlike many memory-saving SNN approaches that trade accuracy/speed/applicability, the proposal targets mathematical equivalence to BPTT (up to numerical backend differences), which is valuable for the community.
- Comprehensive analysis + empirical validation: Reviewers note strong theoretical treatment of memory cost/correctness and extensive experiments across architectures/datasets.
- Rebuttal added concrete profiling and ablations: The authors supplied fine-grained runtime decomposition and isolated ablations of spatial/temporal partitioning and greedy restoration, directly addressing the most substantive reviewer requests.
- Commitment to code release and integration: The authors clearly state an intention to release code upon acceptance and integrate into mainstream SNN frameworks, addressing an important adoption concern.

Weaknesses / remaining concerns
- Online / strict streaming training scope: The method targets layer-wise training (common for modern medium/large SNNs). The rebuttal clarifies partial applicability in online settings and argues memory is less critical there; this is acceptable but should be clearly stated as a limitation.
- Some manual choices remain: Partitioning rules are currently user-specified; authors indicate auto-partitioning as future work. This is fine, but should be clearly positioned.

**Reviewer Scores:**

* **Reviewer pEn5 (score: 8)**
  The reviewer explicitly stated after the rebuttal that the authors clarified all doubts and they will “stick with” the high score.

* **Reviewer eX6a (score: 6)**
  Their main concerns were missing fine-grained runtime profiling and isolated ablations of key components. The authors provided detailed layer-wise runtime decomposition, scaling behavior, and a targeted ablation table disentangling spatial/temporal partitioning and greedy restoration. This directly addresses the core technical weaknesses, so a modest upward revision is plausible.

* **Reviewer XHsK (score: 6)**
  The reviewer’s reservations were mostly about novelty/scope/impact and code release. The rebuttal clarified code release plans and provided systematic comparisons of compression methods; it also argued broader impact (scaling SNNs, longer T, accessibility). These help, but since the reviewer’s critique is partly “conceptual novelty,” they may remain at 6 even if satisfied, with a smaller chance of moving to 7.

* **Reviewer cB7e (score: 2; clearly mismatched review)** **Not meaningful / should be excluded.**
  The review appears to evaluate an LLM memory compression paper rather than this SNN training submission. If the reviewer had participated fully and recognized the mismatch, the correct action would be withdrawal/correction rather than a score adjustment. As-is, it should not be used in scoring aggregation or recommendation.

---

### Decision · Program_Chairs · 2026-01-26

Accept (Poster)